# ANY-SUBGROUP EQUIVARIANT NETWORKS VIA SYMMETRY BREAKING

**Abhinav Goel**[1], **Derek Lim**[1], **Hannah Lawrence**[1], **Stefanie Jegelka**[1,2], **Ningyuan Huang**[3]
[1]MIT, [2]TUM, [3]Flatiron Institute

## ABSTRACT

The inclusion of symmetries as an inductive bias, known as "equivariance", often improves generalization on geometric data (e.g. grids, sets, and graphs). However, equivariant architectures are usually highly constrained, designed for symmetries chosen *a priori*, and not applicable to datasets with other symmetries. This precludes the development of flexible, multi-modal foundation models capable of processing diverse data equivariantly. In this work, we build a single model — the Any-Subgroup Equivariant Network (ASEN) — that can be simultaneously equivariant to several groups, simply by modulating a certain auxiliary input feature. In particular, we start with a fully permutation-equivariant base model, and then obtain subgroup equivariance by using a symmetry-breaking input whose automorphism group is that subgroup. However, finding an input with the desired automorphism group is computationally hard. We overcome this by relaxing from exact to approximate symmetry breaking, leveraging the notion of 2-closure to derive fast algorithms. Theoretically, we show that our subgroup-equivariant networks can simulate equivariant MLPs, and their universality can be guaranteed if the base model is universal. Empirically, we validate our method on symmetry selection for graph and image tasks, as well as multitask and transfer learning for sequence tasks, showing that a single network equivariant to multiple permutation subgroups outperforms both separate equivariant models and a single non-equivariant model.

## 1 INTRODUCTION

Equivariant machine learning exploits symmetries in data to constrain the model with known priors, often leading to improved generalization [Elesedy and Zaidi, 2021; Petrache and Trivedi, 2023], interpretability [Bogatskiy et al., 2024], and efficiency [Bietti et al., 2021]. Most existing equivariant models are tailored to specific symmetry groups chosen *a priori*: for example, graph neural networks (GNNs) and DeepSets with permutation equivariance, convolution neural networks (CNNs) with translation equivariance, and Neural Equivariant Interatomic Potentials satisfying Euclidean group symmetries. While these equivariant models have demonstrated promising performance in datasets satisfying the prescribed symmetries, they are *inflexible* in several important ways: (I) equivariant architectures typically require deriving and implementing group-specific equivariant layers, so substantial research and engineering must be done for architectural design whenever a new type of symmetry arises, and (II) equivariant architectures are typically only equivariant to one symmetry group, and significantly differ across symmetries. Thus, an equivariant model cannot easily transfer knowledge across domains with distinct symmetries, so equivariant models cannot benefit from the empirical successes of the foundation model paradigm [Bommasani et al., 2021].

In this work, we introduce a framework for building *flexible* equivariant networks, named *Any-Subgroup Equivariant Networks* (ASEN). Given a base group $\mathbf{G}$, we consider its subgroups $G \leq \mathbf{G}$ (typically known a priori) that capture the symmetries intrinsic to the domain (e.g. graph automorphisms) or induced by the task (e.g., sequence reversal). To design a subgroup-equivariant model $f$, we start with a base network $h_\theta$, which is equivariant to the large group $\mathbf{G}$ and thus overly constrained for our purpose — it cannot represent functions that are equivariant only to $G$ but not $\mathbf{G} \setminus G$. To reduce the amount of constraints, we augment the input $x$ with a feature $\mathbf{v}$ that break the symmetries in $\mathbf{G} \setminus G$ but maintain the symmetries in $G$. To ensure this, we construct $\mathbf{v}$ so that its self-symmetry group (i.e. automorphism group) is equal to $G$, i.e. $\mathrm{Aut}(\mathbf{v}) = G$. Finally, we pass in the symmetry-breaking input $\mathbf{v}$ to our overly-constrained $h_\theta$, and obtain the model $f_\theta(x) = h_\theta(x, \mathbf{v})$. We prove that the

model $f_\theta$ is indeed $G$-equivariant, and under certain conditions it is not equivariant to $\mathbf{G} \setminus G$; in other words, it has the correct equivariance. A trivial, but widespread, example of this technique is the use of positional encodings [Vaswani et al., 2017] to fully break the permutational symmetry of transformers (since every entry of the sinusoidal positional encoding vector is unique, $G = \mathrm{Aut}(\mathbf{v})$ is the trivial group). When $\mathbf{v}$ has non-trivial automorphism, *some* equivariance is retained.

ASEN overcomes inflexibility (I), since it only requires providing a single new input $\mathbf{v}$ (with correct automorphism group) to a base network $h_\theta$. Also, ASEN overcomes inflexibility (II), since a single instance of our model can process data from varying domains with different symmetry groups.

While our framework works for any $\mathbf{G}$, we focus on the particular case when $\mathbf{G} = S_n$ is the symmetry group acting as permutation matrices, so that our networks are equivariant to permutation subgroups. This covers the symmetries of many common domains such as sets and graphs, which allows us to leverage existing permutation equivariant models as the base model $h_\theta$. In particular, we may leverage existing set networks (inputs in $\mathbb{R}^n$), graph neural networks (inputs in $\mathbb{R}^{n^2}$) and hypergraph neural networks (inputs in $\mathbb{R}^{n^K}$) for the base model. While large $K$ may be required for complex groups, to balance efficiency and expressivity, we focus on the $K = 2$ case of graph neural networks, and develop a practical algorithm for computing the symmetry breaking object $\mathbf{v}$ as edge features, with (nearly) the desired self-symmetry $\mathrm{Aut}(\mathbf{v}) \approx G$. To do this, we use the notion of the 2-closure $G^{(2)}$ of a group $G$ [Ponomarenko and Vasil'ev, 2020], which provides a formal notion of a group that is close to the target group ($G^{(2)} \approx G$), and we compute $\mathbf{v}$ with $\mathrm{Aut}(\mathbf{v}) = G^{(2)}$.

Theoretically, we show that under mild conditions, ASEN parameterized with graph neural networks has the exact permutation subgroup equivariance. Further, we prove that ASEN is expressive in two senses: it can approximate certain equivariant MLPs [Maron et al., 2019; Finzi et al., 2021b] to arbitrary accuracy, and it is universal in the space of $G$-equivariant functions if the base model is universal over $\mathbf{G}$-equivariant functions.

To validate our approach, we apply ASEN to diverse settings: (1) exploiting symmetries *within a single task*, including graph learning (human pose estimation and traffic flow prediction) and image classification (Pathfinder); (2) leveraging symmetries *across different tasks* on sequences in multitask and transfer learning. Across these settings, our results highlight the flexibility of ASEN and practical utility of symmetry-aware architectures. Our framework supports both fine-grained control over group actions and strong transfer of learned representations, making it a powerful tool for structured generalization in neural networks. Our main contributions are summarized as follows:

- We propose Any-Subgroup Equivariant Networks (ASEN), a framework for building a flexible equivariant model capable of modeling distinct symmetries across diverse tasks.

- We theoretically show that ASEN enforces any desired subgroup symmetry via proper choice of the symmetry-breaking input and architecture. We also prove that ASEN is as expressive as equivariant MLPs, with universality guarantees given a sufficiently expressive base model.

- We validate our framework in applications including symmetry selection, multitask learning and transfer learning, highlighting its flexibility and effectiveness in exploiting shared symmetry structures.

## 2 RELATED WORK

**Subgroup Equivariance** In equivariant network design, recent works [Blum-Smith et al., 2025; Ashman et al., 2024; Lim et al., 2024] have proposed subgroup-equivariant models via augmenting an auxiliary input. Specifically, Blum-Smith et al. [2025] proposed a permutation-invariant model for symmetric matrices by using a DeepSet base model—invariant to a bigger group, together with a suitable symmetry-breaking parameter to reduce the base model symmetries; Ashman et al. [2024] used fixed symmetry breaking inputs to construct non-equivariant models or approximately equivariant ones; Lim et al. [2024] leveraged node or edge features to reduce the amount of permutation symmetries in neural network computational graphs. While these works model subgroup equivariance, they are limited to the single-task setting. In contrast, we study the multitask and transfer learning setting, providing a recipe to build flexible equivariant models.

**Symmetry Breaking** Symmetry breaking via node identification is a popular technique to enhance the expressivity of graph neural networks [Abboud et al., 2021; Sato et al., 2021; Bevilacqua et al., 2025]. Symmetry breaking of the *input* has also been used to improve the flexibility of equivariant models for applications in graph generation and physical modeling [Smidt et al., 2021; Lawrence et al., 2024; Xie and Smidt, 2024]. Unlike [Smidt et al., 2021; Lawrence et al., 2024; Xie and Smidt, 2024] that perform input-dependent symmetry breaking, we break the symmetry of the *model* uniformly for all inputs. Moreover, existing works typically focus on (approximate) equivariance to one particular group. In contrast, we build a single model capable of modeling diverse data equivariantly, via different choices of $\mathbf{v}$ adapted to the target application.

**Approximate and Adaptive Equivariance** Another direction towards flexible equivariant networks relies on approximate equivariance [Wang et al., 2022; Huang et al., 2023], soft equivariance by converting architectural constraints into a prior [Benton et al., 2020; Finzi et al., 2021a], regularization [Kim et al., 2023], or adaptive equivariance per task and environment [Gupta et al., 2024]. Approximate equivariance can also improve generalization [Wang et al., 2022; Huang et al., 2023]. This motivates our approach that approximates the automorphism group of the symmetry breaking input with the 2-closure group.

## 3 METHOD

### 3.1 GENERAL METHOD (ASEN)

Here, we describe our general framework for ASEN. Let $\mathbf{G}$ be a matrix group, and let $G \leq \mathbf{G}$ be a subgroup. Both groups have actions on the sets $\mathcal{X}$ and $\mathcal{Y}$. We desire our model to parameterize $G$-equivariant functions, that is, functions $f : \mathcal{X} \to \mathcal{Y}$ such that $f(gx) = gf(y)$ for $g \in G$. To this end, we consider a "lift" of $f$: a function $h_\theta : \mathcal{X} \times \mathcal{V} \to \mathcal{Y}$ on an expanded space, where we introduce an additional space $\mathcal{V}$ on which $G$ and $\mathbf{G}$ act. The function $h_\theta$ is $\mathbf{G}$-equivariant (i.e. equivariant to the larger group):

$$h_\theta(gx, gv) = gh_\theta(x, v), \quad \forall g \in \mathbf{G}. \tag{1}$$

To obtain $f_\theta$ from $h_\theta$, we find a symmetry-breaking input $\mathbf{v} \in \mathcal{V}$ that is exactly self-symmetric to the subgroup $G$, i.e. its automorphism group is $G$,

$$\mathrm{Aut}(\mathbf{v}) = \{g \in \mathbf{G} : g\mathbf{v} = \mathbf{v}\} = G. \tag{2}$$

Finally, we define our $G$-equivariant model $f_\theta : \mathcal{X} \to \mathcal{Y}$ as

$$f_\theta(x) = h_\theta(x, \mathbf{v}). \tag{3}$$

The model $f_\theta$ is $G$-equivariant because for any $g \in G$,

$$f_\theta(gx) = h_\theta(gx, \mathbf{v}) = h_\theta(gx, g\mathbf{v}) = gh_\theta(x, \mathbf{v}) = gf_\theta(x), \tag{4}$$

where the second equality follows from $g \in \mathrm{Aut}(\mathbf{v})$, and the third equality is due to $\mathbf{G}$-equivariance of $h_\theta$. For $g \in \mathbf{G} \setminus G$, this equality can break: if $g\mathbf{v} \neq \mathbf{v}$, then $h_\theta(gx, \mathbf{v}) \neq h_\theta(gx, g\mathbf{v})$. In fact, if $h_\theta$ is injective for the input $\mathbf{v}$, then $f_\theta$ is *only* equivariant to $G$, and not to any other elements in the larger group $\mathbf{G}$. These results are captured in Prop. 1. As an example: consider $\mathbf{G} = O(3)$ acting on 3D point cloud $x \in \mathbb{R}^{n \times 3}$, and the $O(3)$-equivariant base model $h_\theta$. Now we fix a particular axis $\mathbf{v}$ with the stabilizer $O(2)$. If $h_\theta$ is injective for $\mathbf{v}$, then $f_\theta = h_\theta(x, \mathbf{v})$ is only equivariant to $O(2)$ but not $O(3) \setminus O(2)$.

### 3.2 PERMUTATION SUBGROUP EQUIVARIANCE VIA HYPERGRAPH SYMMETRY BREAKING

To use our ASEN framework for parameterizing a $G$-equivariant function, we need two main components: (i) a method of parameterizing the base model $h_\theta$ that is equivariant to the larger group $\mathbf{G}$, and (ii) a way to construct or compute a symmetry breaking object $\mathbf{v}$ with automorphism group $\mathrm{Aut}(\mathbf{v}) = G$. In this subsection, we show that when $\mathbf{G} = S_n$ is the symmetric group acting as permutation matrices on $n$ objects, we can leverage existing equivariant architectures for (i); and we can develop a practical algorithm for (ii). For the rest of this paper, we primarily focus on this setting.

To construct efficient and expressive symmetry breaking objects, we turn to *hypergraphs*. Concretely, for a (matrix) group $G$ acting on $\mathbb{R}^n$, a hypergraph on $n$ nodes is defined as $\mathcal{H} =$

$(A^{(1)}, A^{(2)}, \dots, A^{(K)})$, where $A^{(k)} \in \mathbb{R}^{n^k}$ is an order $k$-tensor, and $K$ is the max tensor order. We can interpret $A^{(1)}$ as a (node) positional encoding, and $A^{(k)}$ as (hyper-)edge features for $k \geq 2$. The *automorphism group* of the hypergraph is defined as

$$\text{Aut}(\mathcal{H}) = \{P \in S_n : P^{\otimes^k} A^{(k)} = A^{(k)}, k = 1, \dots, K\}. \tag{5}$$

For instance, if $K = 2$, then this is the standard graph automorphism group

$$\text{Aut}(\mathcal{H}) = \{P \in S_n : PA^{(1)} = A^{(1)}, PA^{(2)} P^\top = A^{(2)}\}. \tag{6}$$

For $K$ large enough, we can construct a hypergraph $\mathcal{H}$ such that $\text{Aut}(\mathcal{H})$ uniquely determines $G$ [Wielandt, 1969]. Then, we consider any existing permutation-equivariant hypergraph neural network $h_\theta : \mathbb{R}^n \times \prod_{k=1}^K \mathbb{R}^{n^k} \to \mathbb{R}^n$ such that any $P \in S_n$,

$$h_\theta(PX, PA^{(1)}, \dots, P^{\otimes^K} A^{(K)}) = Ph_\theta(X, A^{(1)}, \dots, A^{(K)}). \tag{7}$$

We abbreviate (7) by $h_\theta(P(X, \mathcal{H})) = Ph_\theta(X, \mathcal{H})$. Our $G$-equivariant model $f$ then takes the form

$$f_\theta(X) = h_\theta(X, A^{(1)}, \dots, A^{(K)}) \equiv h_\theta(X, \mathcal{H}). \tag{8}$$

**Hypergraph Construction and Approximation with 2-Closure**   Achieving exact symmetry breaking of $S_n$ to the desired subgroup $G$ may require a hypergraph $\mathcal{H}$ of prohibitively high order (up to $K \leq n$). For efficiency, we fix $K = 2$ and construct positional and edge features $\mathcal{H} = (A^{(1)}, A^{(2)})$, whose automorphism group $\text{Aut}(\mathcal{H})$ reflects on how $G$ acts on nodes and pairs of nodes. Concretely, nodes $i, j$ (or node pairs $(i_1, i_2), (j_1, j_2)$) are assigned the same feature if and only if they are in the same $G$-orbit. In this way, the positional and edge features encode the orbit partition under $G$. By construction, $\text{Aut}(A^{(2)})$ is the 2-closure group of $G$, denoted as $G^{(2)}$ [Ponomarenko and Vasil'ev, 2020]. In general $G \leq G^{(2)}$, and in many cases $G = G^{(2)}$ regardless of the permutation representations; such groups are called *totally 2-closed*. This class includes finite nilpotent groups that are either cyclic or a direct product of a generalized quaternion group with a cyclic group of odd order [Abdollahi and Arezoomand, 2018]. See Sec. 5 for concrete examples. When $G < G^{(2)}$, the 2-closure introduces additional symmetries, leading to a symmetry group mismatch. We can use results from [Huang et al., 2023] to analyze the approximation-generalization tradeoff.

Algorithm 1 provides the procedure to compute $A^{(2)}$ with high-level SymPy commands, assuming access to the generating elements of $G$ [1]; see Fig. 1 for examples and App. A for details. We remark that while Alg. 1 computes all pairwise edge features, it can be applied to a sparse graph or a subset of edge features by restricting the support of $\mathcal{H}$.

---

**Algorithm 1** Compute Edge Orbits $A^{(2)}$ such that $\text{Aut}(A^{(2)}) = G^{(2)}$ (SymPy commands)

---

**Require:** Generators $\sigma_1, \dots, \sigma_r$ of $G \leq S_n$
**Ensure:** Edge orbits $A^{(2)} \in [n] \times [n]$ where $A_{ij}^{(2)} = A_{mn}^{(2)} \iff (i, j) \sim_G (m, n)$.
 1: **Lift generators:** For each $\sigma_i \in S_n$, define $\rho_i \in S_{n^2} : (x_a, x_b) \mapsto (\sigma_i(x_a), \sigma_i(x_b))$.
       Encode $(x_a, x_b)$ as $a \cdot n + b$ and construct $\rho_i$ as `Permutation` of size $n^2$.
 2: **Form diagonal subgroup:** Let $\Delta(G) := \langle \rho_1, \dots, \rho_r \rangle$ be the subgroup of $S_{n^2}$.
       `Delta = PermutationGroup([ρ_1,...,ρ_r]).`
 3: **Compute edge orbits:** For an edge $(x_a, x_b)$, apply $\rho_i$ repeatedly until no new pairs can be found.
       `Delta.orbits()`

---

**Alignment**   Note that it is important that the inputs have their node indices aligned to each other, i.e. they share a common labeling. This is implicitly captured via the choice of the concrete matrix group $G$. For example, the inputs in Fig. 1 are treated as sequences (instead of sets). The alignment is typically satisfied in applications such as sequence modelling, graph signal processing, and graph time series (see examples in Sec. 5), but needs to be computed for other applications such as graph-level tasks.

---

[1] If instead we are given *all* elements of $G$, we can pass them to `PermutationGroup` in SymPy and run the Schreier–Sims algorithm. This produces a *base and strong generating set (BSGS)*: a compact, non-redundant set of generators adapted to a stabilizer chain. The BSGS allows efficient orbit and membership computations without ever enumerating the full group.

Figure 1: Example symmetry breaking objects as positional features $A^{(1)}$ and edge features $A^{(2)}$ for encoding subgroup symmetries in 4-node paths. These symmetries are explored further in Sec. 5.

## 4 THEORETICAL RESULTS

We first establish that under mild conditions, ASEN with exact symmetry breaking can achieve the desired subgroup symmetry $G \leq \mathbf{G}$ for any general matrix group $\mathbf{G}$ (Prop. 1), and for the permutation group $S_n$ in the context of graph learning (Lem. 1). We then show that ASEN with approximate symmetry breaking can simulate equivariant MLPs (Thm. 1), and enjoys universality guarantees if the base model is universal (Thm. 2). Proofs are deferred to App. B.

The following Prop. 1 shows that ASEN has precisely the desired equivariance to $G$ under mild conditions.

**Proposition 1.** *Let $h_\theta : \mathcal{X} \times \mathcal{V} \to \mathcal{Y}$ be $\mathbf{G}$-equivariant, and let $\mathrm{Aut}(\mathbf{v}) = G$. Then $f_\theta(x) := h_\theta(x, \mathbf{v})$ is equivariant to $G$. If additionally $h_\theta$ is injective in the input $\mathbf{v}$, then $f_\theta$ is not equivariant to any transformation in $\mathbf{G} \setminus G$.*

**Hypergraph Symmetry Breaking**    As noted in Section 3.2, we generally take $h_\theta$ to be a graph neural network when parameterizing permutation subgroup equivariant functions. In what follows, we characterize the condition for correct equivariance (i.e., $h_\theta$ is equivariant to $G$ but not $\mathbf{G} \setminus G$) when $h_\theta$ is a one-layer message-passing neural network (MPNN) and $\mathbf{G} = S_n$ operating on nodes and edges, and $G = \mathrm{Aut}(A^{(2)})$ is the automorphism group of a weighted graph $A^{(2)} \in \mathbb{R}^{n \times n}$. By the definition of message-passing, the output $h_\theta$ at node $i$ is computed as

$$h_\theta(X, A^{(2)})[i] = \phi \left( \psi_n(X_i), \tau \left( \{\!\!\{ \psi_e(X_i, X_j, A^{(2)}_{i,j}) \mid j \in [n] \}\!\!\} \right) \right) \tag{9}$$

where $\psi_e$ is the edge update function, $\psi_n, \phi$ are the node update functions, and $\tau$ is the edge multiset aggregation. In the following lemma, we show that if the constituent functions are injective, then $h_\theta$ is correctly equivariant to only $G$ under mild assumptions on the node features.

**Lemma 1.** *If a one-layer MPNN $h_\theta$ uses injective functions for (hyper-)edge feature update $\psi_e$, node update $\phi$, and (hyper-)edge multiset aggregation $\tau$, and if the node features are distinct, then $h_\theta$ is not equivariant to permutations in $S_n \setminus G$ where $G = \mathrm{Aut}(A^{(2)})$.*

Note that these injectivity conditions are similar to sufficient conditions under which message passing graph neural networks have the same expressive power as the 1-Weisfeiler-Leman graph isomorphism test [Xu et al., 2019]. We can similarly conclude that these conditions are sufficient for deeper MPNNs to have the correct $G$-equivariance. Also, we can show an analogous result for hypergraph networks by viewing them as higher-order MPNNs operating on hyperedges, where $\sigma \in S_n \setminus G$ implies that there exists a hyperedge $(i, j, \ldots, k)$ such that $A_{i,j,\ldots,k} \neq A_{\sigma(i),\sigma(j),\ldots,\sigma(k)}$.

**Connections to Equivariant MLPs**    Here, we show that ASEN with approximate symmetry breaking can simulate certain configurations of a common type of equivariant neural network, sometimes called an *equivariant MLP*, which consists of equivariant linear maps and elementwise nonlinearities [Ravanbakhsh et al., 2017; Maron et al., 2019; Finzi et al., 2021b]. When applied to a group $G$ that acts as permutation matrices on $\mathbb{R}^n$, an equivariant MLP is defined as a composition: $T_L \circ \sigma \circ \cdots \circ \sigma \circ T_1$, where $\sigma$ is an elementwise nonlinearity, and each $T_i : \mathbb{R}^{n^{k_i}} \to \mathbb{R}^{n^{k_{i+1}}}$ is a $G$-equivariant linear map (for simplicity, we ignore channel dimension here). We call $k^* = \max_i k_i$ the *order* of the G-MLP, so if $T_i : \mathbb{R}^n \to \mathbb{R}^n$ for each $i$ then the G-MLP has order 1. We prove the following result:

**Theorem 1.** *Any order 1 G-MLP can be approximated to arbitrary accuracy on a compact domain via ASEN with $K = 2$ and an MPNN base model $h_\theta$.*

We remark that much like an equivariant MLP can increase expressivity by increasing the order $k^*$, we can increase expressivity in ASEN by increasing $K \geq 2$ and using the $K$-closure group approximation [Ponomarenko and Vasil'ev, 2020]. While we show this relationship in expressivity at $k^* = 1$ and $K = 2$, we believe that there may be relationships between the two methods at higher orders, and leave it to future work.

**Universality Results** We next show that the universality of ASEN follows from the universality of its base model.

**Theorem 2.** *Let $\mathbf{G}$ be a compact group, $\mathcal{X}, \mathcal{V}$ be compact metric $\mathbf{G}$-spaces, and $\mathcal{Y}$ be a compact $\mathbf{G}$-space. Let $f_\theta : \mathcal{X} \times \mathcal{V} \to \mathcal{Y}$ be a universal family of continuous $\mathbf{G}$-equivariant networks, i.e. $f_\theta(gx, gv) = g \cdot f_\theta(x, v)$. Consider $\mathcal{H} \in \mathcal{V}$ with stabilizer equal to a subgroup $G \leq \mathbf{G}$. Then, the family $f_\theta(\cdot, \mathcal{H})$ is universal over continuous $G$-equivariant functions from $\mathcal{X}$ to $\mathcal{Y}$.*

We remark that there exist universal architectures for point clouds [Hordan et al., 2024; Blum-Smith et al., 2025]. For graphs, universal architectures typically require a combinatorially large feature space [Maron et al., 2019], or only hold in a probabilistic sense [Abboud et al., 2021]. However, if we restrict to graphs with unique node features, then MPNNs (with sufficient depth and width) are universal [Loukas, 2020]. Consequently, Thm. 2 ensures that whatever variety of universality the family $f_\theta$ enjoys is inherited by $f_\theta(\cdot, \mathcal{H})$.

## 5 EXPERIMENTS

Towards developing a general-purpose equivariant foundation model, we evaluate ASEN in diverse experimental settings [2] by answering the following questions:

- Q1 For a single task, can ASEN —with one architecture—explore different symmetries and reveal the impact of the group choice (Sec. 5.1)?

- Q2 Can ASEN leverage shared symmetry structure across tasks to outperform task-specific equivariant models or non-equivariant baselines in multitask learning (Sec. 5.2.1) and transfer learning (Sec. 5.2.2)?

**Architecture** Our backbone is a permutation-equivariant graph neural network (GNN), composed of input layers (e.g. embedding, MLP), followed by four layers of GATv2 message-passing [Veličković et al., 2017], and concluded with output layers (e.g., projection, aggregation). Standard dropout and layer normalization are applied throughout. There are two task-specific modules: *EdgeEmbedder* that calls Alg. 1 to categorize edge orbits and *learn their embeddings*; *TokenEmbedder* that maps (discrete) node features to learnable token embeddings for classification tasks (omitted for regression tasks); See Fig. 2 for an overview. The preprocessing cost (Alg. 1) scales as $O(rn^2)$ where $n$ is the number of nodes and $r$ is the number of the generators of the group $G$. The training cost, using a message-passing GNN backbone, scales as $O(mn^2)$ on a fully-connected dense graph where $m$ is the number of training data points, and $O(m|E|)$ on a sparse graph where $|E|$ is the number of edges.

As *EdgeEmbedder* is a learnable module, ASEN can *discover more symmetries* from data if the chosen group $G^{(2)}$ (specifying the edge orbits) is smaller than the target group; see evidence in Sec. 5.2. On the other hand, ASEN can fail when $G^{(2)}$ is much larger than $G$, as we show in App. C.5.

### 5.1 SYMMETRY MODEL SELECTION APPLICATIONS

In this section, we consider exploring different symmetry choices for a given task. Specifically, for each subgroup $G$ in a candidate set chosen a priori, we train a new instance of ASEN with edge features satisfying $\text{Aut}(A^{(2)}) = G^{(2)}$ (or positional features satisfying $\text{Aut}(A^{(1)}) = G^{(1)}$). Our results highlight the utility of choosing subgroup symmetries informed by the structure of the domain,

---

[2]Source code available at `https://github.com/amgoel21/perm_equivariance_graph_formulation`.

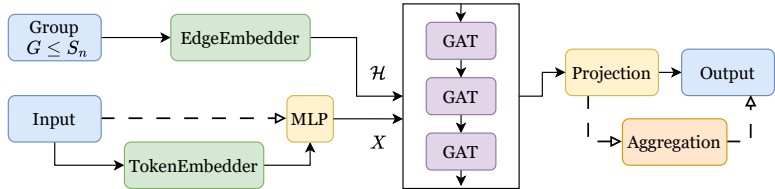

Figure 2: ASEN Architecture to model any permutation subgroup-equivariant functions.

such as the reflection symmetry in skeleton graphs, the continuous road structure in traffic graphs, and the local symmetry in image grids.

### 5.1.1 GRAPH TASKS

We apply ASEN to perform symmetry model selection for learning on a fixed graph setting, using the experimental set-up from Huang et al. [2023]. Unlike Huang et al. [2023] that constructs distinct $G$-equivariant MLPs for each group, ASEN offers a unified architecture to flexibly model different $G$-equivariance by symmetry breaking (i.e., via positional or edge features).

**Human Pose Estimation** We begin with an application in human pose estimation, using the Human3.6M dataset [Ionescu et al., 2014], which consists of 3.6 million human poses from various images. Our input features consist of 2D coordinates $X \in \mathbb{R}^{16 \times 2}$ representing joint positions on a skeleton graph $A \in \{0, 1\}^{16 \times 16}$ (see Figure inset). The model predicts the corresponding 3D joint positions in $\mathbb{R}^{16 \times 3}$. Performance is evaluated using the standard P-MPJPE (Procrustes-aligned Mean Per Joint Position Error) metric. 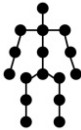 We consider three kinds of edge feature $\mathcal{H} = A^{(2)}$ (Alg. 1): (1) fully-connected $A_f^{(2)}$, (2) sparse $A_s^{(2)} \equiv A$ where edges are constrained to the support of the skeleton graph $A$, and (3) weakly sparse combining $A_f^{(2)} + A_s^{(2)}$. We consider *selecting* different automorphism groups of the human skeleton edges $\mathrm{Aut}(A^{(2)})$ (that yields the best performance): $S_2$ (full left-right reflection), $S_2^2$ (left arm/right arm and left leg/right leg), $S_2^6$ (each left-side joint independently mapped to corresponding joint on right side), and $I$ (no equivariance). Our results in Tab. 1 obtained from a single model ASEN match with those reported in [Huang et al., 2023] that require multiple distinct equivariant MLPs. Notably, the weakly sparse graph yields some of the strongest results, highlighting ASEN's flexibility in capturing multiple symmetries. This approach can also be extended to represent a graph with multiple sets of symmetries.

Table 1: P-MPJPE error ($\downarrow$) for human pose estimation using different symmetry groups and edge frameworks.

| Group | Fully Connected | Sparse | Weakly Sparse |
|---|---|---|---|
| $I$ | 34.71 | **33.39** | 34.75 |
| $S_2$ | 39.48 | 40.52 | **38.80** |
| $S_2^2$ | 43.24 | 42.37 | **40.67** |
| $S_2^6$ | 47.54 | 49.45 | **46.52** |

Table 2: Mean Absolute Error (MAE $\downarrow$) for traffic flow prediction.

| Model, Group ($\sum_i n_i = n$) | MAE |
|---|---|
| Fully Connected, $S_{n_1} \cdot S_{n_2}$ | 2.72 |
| Sparse, $S_{n_1} \cdot S_{n_2}$ | **2.69** |
| Fully Connected, $S_{n_1} \cdot \cdots \cdot S_{n_9}$ | 2.79 |
| Sparse, $S_{n_1} \cdot \cdots \cdot S_{n_9}$ | 2.77 |
| DCRNN [Li et al., 2018], $S_n$ | 2.77 |

**Traffic Prediction** Next, we evaluate on the traffic forecasting task, also taken from [Huang et al., 2023]. This uses the METR-LA dataset, containing time-series traffic data from 207 sensors deployed on Los Angeles highways. The sensor (node) records speeds and traffic volume every 5 minutes, resulting in a graph time series with node feature $X_t \in \mathbb{R}^{207 \times 2}$. The objective is to predict traffic conditions at future time $X_{t+1}$ based on the past traffic $\{X_t, X_{t-1}, X_{t-2}\}$. The underlying graph is defined via a sensor adjacency matrix $A \in \mathbb{R}^{207 \times 207}$ constructed from roadway connectivity.

To incorporate symmetry structure, we leverage the spatial layout of sensors along major highways. We consider two group structures, taken from [Huang et al., 2023]: one with two clusters representing

Table 3: Examples of synthetic tasks: input, equivariant and invariant output, and the target group.

| Task | Input | Equivariant Output | Invariant Output | Group $G \le S_n$ |
|---|---|---|---|---|
| Intersect | $[a, b, c, \mid b, c, d]$ | $[0, 1, 1, \mid 1, 1, 0]$ | 2 | $S_{n/2} \times S_{n/2} \times S_2$ |
| Symmetric Difference | $[a, b, c, \mid b, c, d]$ | $[1, 0, 0, \mid 0, 0, 1]$ | 1 | |
| Palindrome ($k = 3$) | $[a, b, c, b, d]$ | $[0, 1, 1, 1, 0]$ | True | Sequence Reversal |
| Monotone Run ($k = 3$) | $[3, 1, 2, 4, 1]$ | $[0, 1, 1, 1, 0]$ | True | |
| Cyclic Sum ($c = 3$) | $[7, 1, 2, 9, 8]$ | $[1, 0, 0, 1, 1]$ | 24 | $C_n$ (cyclic shifts) |
| Cyclic Product ($c = 3$) | $[1, 2, 4, 0, 5]$ | $[1, 1, 0, 0, 1]$ | 10 | |
| Detect Capital | $[A, b, b, b]$ | – | True | $S_{n-1}$ (permute all except first) |
| Longest Palindrome | $[a, b, c, c, c, c, d, d]$ | – | 7 | $S_n$ |

major highway branches $G = S_{n_1} \times S_{n_2}$, and another with nine clusters corresponding to finer-grained regional groupings $G = S_{n_1} \times \ldots \times S_{n_9}$. These symmetry groups serve as approximate equivariances, which encourage learning equivariant representations for similarly situated sensors. We construct node positional features such that $\text{Aut}(A^{(1)}) = G$, and benchmark our model in both fully-connected and sparse graph regimes. As shown in Tab. 2, choosing suitably smaller symmetry can outperform full permutation symmetry.

### 5.1.2 Pathfinder Task

As discussed in Sec. 1, Transformer with PE (1D or 2D) breaks permutation symmetry completely, since each position feature is unique (i.e., trivial automorphism). We study the effect of imposing local symmetry on Transformers for image tasks, by sharing the same position vector for pixels within the same $p \times p$ patch for $p = 2, 3, 4$. This patch-wise weight sharing preserves permutation symmetry within patches while distinguishing different patches. We evaluate these 2D-PE variants of Transformer on the Pathfinder-64 task [Tay et al., 2021] using a standard architecture with learnable row/column embeddings. Fig. 3 shows that using local permutation symmetry can improve performance while slightly reducing the model parameter counts (see App. C.1 for details).



Figure 3: Pairwise distances of learned positional encodings of Transformer on Pathfinder-64, with test accuracy shown at the top: imposing local permutation symmetry improves performance.

### 5.2 Synthetic Sequence Modeling Tasks

To assess the ability of ASEN to transfer structural knowledge *across tasks*, we consider synthetic sequence modelling tasks capturing various permutation subgroup symmetries, summarized with examples in Tab. 3 (with more details deferred to Tab. 5). To demonstrate these sequence tasks can benefit from exploiting subgroup symmetries, we compare two ASEN variants: one *equivariant* model with the *correct* group (c.f. rightmost column in Tab. 3), and one *non-equivariant* baseline without any symmetry. As shown in Fig. 4, equivariant models incorporating the correct symmetry significantly outperform their non-equivariant counterparts across all tasks. More details including similar results for the invariant setting (see Tab. 6) and the effect of varying training dataset size (see Fig. 9) can be found in App. C.2.

Additionally, we explore if ASEN can learn more symmetries from data given a misspecified symmetry group (smaller than the target group). We consider the Intersect task with the target group $G = (S_{n/2})^2 \times S_2$, but only encode a smaller symmetry group $G' = (S_{n/2})^2$ in the edge features. We check the edge feature weights before and after training to see if the $S_2$ symmetry was learned.

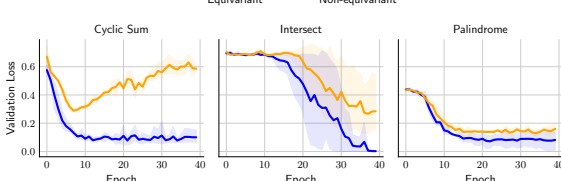
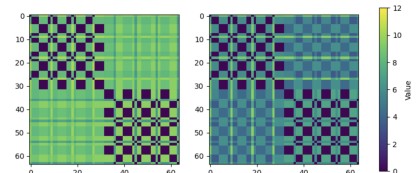

Figure 4: ASEN with the correct group ("Equivariant") converges faster and to a lower loss than its trivial symmetry counterpart ("Non-equivariant").

Figure 5: Initial edge weights (left) and trained weights (right): ASEN learns more symmetries from data.

Fig. 5 shows pairwise distances between the learned Intersect edge weights: the top-left and bottom-right quadrants show the correct structure (checkerboard), whereas the top-right and bottom-left quadrants lack this initially but converge to the checkerboard pattern after training, discovering the $S_2$ symmetries from data.

### 5.2.1 MULTITASK LEARNING APPLICATIONS

In this section, we investigate whether learning related tasks with compatible or similar symmetry groups can benefit ASEN from shared representation learning. We again make use of the synthetic tasks introduced in Tab. 3. While each task can be learned independently using a different equivariant model, we assess whether *multitask training* can facilitate improved generalization in low-data regimes. To this end, we use the same ASEN backbone (including weights) shared across tasks, while the TokenEmbedder and EdgeEmbedder modules are task-specific. During training, we randomly sample batches from all tasks, ensuring concurrent and balanced updates across tasks.

We then compare two regimes: one training only on $r$ units ("1 unit" represents 2,500 datapoints) of a *single-task*, and the other training on $r$ units with $n_{\text{task}} = 3$ tasks, specifically Intersect, Cyclic Sum, and Palindrome (see Tab. 3). We vary $r \in \{0.2, 0.4, 0.6, 0.8, 1.0, 2.0, 3.0\}$ and report the average performance across three random seeds. Fig. 6 (top) shows that multitask training leads to significantly improved convergence and test accuracy in the low-data setting for learning Intersect, demonstrating the benefit of symmetry-aligned task transfer *on certain tasks*; on the other tasks (Cyclic Sum, Palindrome), the multitask setting did not provide much benefit (see details in App. C.3).

To further investigate the effect of the number of tasks, we repeat the above multitask training with $n_{\text{task}} \in \{4, 5, 6\}$ tasks, by adding Symmetric Difference, Cyclic Product, and Monotone Run, respectively (see Tab. 3). Fig. 6 (bottom) shows that while increasing $n_{\text{task}}$ improves performance in low-data regimes (e.g., $r \leq 1.0$ units), such effect diminishes as training size grows. As the compute cost scales linearly with both training set size and the number of tasks (see Fig. 10 for details), this highlights a practical tradeoff in choosing between training size and task count.

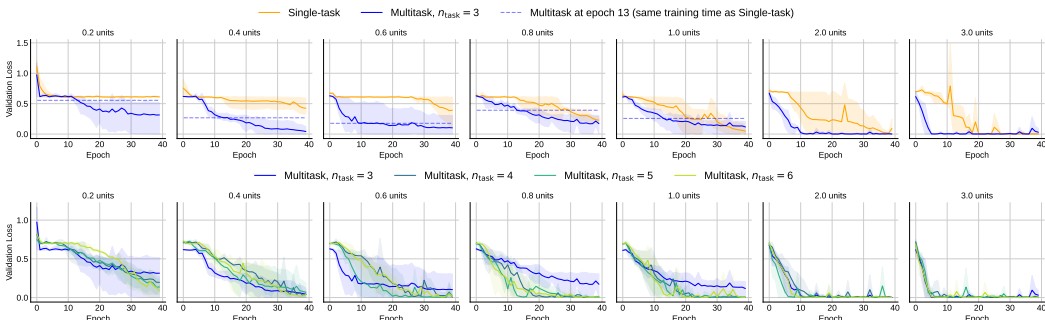

Figure 6: (Top) Multitask versus single-task performance on Task Intersect with varying training set sizes ($r$ units), showing the benefit of multitask training particularly on low-data regimes, even when matched to the same total training time as the single-task setting (dashed horizontal lines); (Bottom) Multitask performance with varying number of tasks ($n_{\text{task}}$), showing the diminishing returns of adding more tasks as the training size increases.

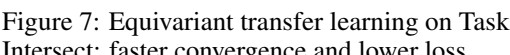
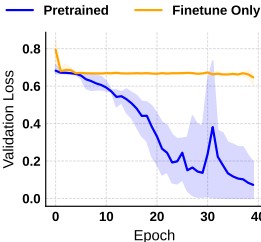
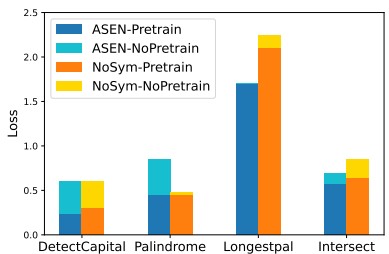

Figure 7: Equivariant transfer learning on Task Intersect: faster convergence and lower loss.

Figure 8: Invariant transfer learning: improved pretraining when using symmetry in ASEN.

### 5.2.2 Transfer Learning Applications

Beyond multitask training, we explore transfer learning by pretraining on tasks with diverse symmetries and then finetuning on a new task with a distinct symmetry. We assume knowledge of the task domain to leverage and transfer the domain-related symmetries (i.e., knowing the base group $\mathbf{G}$ covering most downstream task symmetries as its subgroups). We focus on the same set of synthetic tasks in Sec. 5.2.1 and simulate a low-resource regime by limiting access to only 0.15 units (375 datapoints) of training data. We compare two ASEN model variants: A *finetune-only* baseline trained from scratch using only the chosen task data, and a *pretrained* model initialized via joint training on 1.5 units each of the other tasks, followed by fine-tuning on the chosen task data. To encourage knowledge retention, we reduce the learning rate of the GNN backbone during fine-tuning, while allowing the embedding layers to update more freely. For the equivariant setting, Fig. 7 shows that pretraining ASEN achieves significantly better generalization compared to training from scratch on the Intersect task, showcasing ASEN as an effective initialization for related downstream tasks. For the invariant setting, Fig. 8 shows that pretraining ASEN with the correctly specified symmetry outperforms its trivial symmetry baseline, and pretraining has a larger transfer effect when symmetry is provided on the Detect Capital and Palindrome tasks. Ablation studies on the pretraining effect with respect to varying sequence length are summarized in Fig. 11 and discussed in App. C.4.

## 6 Conclusion and Future Directions

In this work, we introduce ASEN, a framework for building a flexible equivariant model capable of exploiting diverse symmetries. Given a subgroup $G \leq \mathbf{G}$, ASEN parameterizes $G$-equivariant functions via a base $\mathbf{G}$-equivariant model, and a symmetry breaking object whose autormophism group is $G$. For general $\mathbf{G}$, we prove that ASEN can achieve the desired subgroup symmetry and inherit universality properties from the base model. Focusing on the permutation group $\mathbf{G} = S_n$, we encode the desired subgroup symmetries via positional and edge features, which can be easily integrated to standard GNNs and expressive as equivariant MLPs. We empirically demonstrate the flexibility of ASEN to perform symmetry selection in graph and image tasks, and its effectiveness in exploiting shared symmetry structures in multitask and transfer learning for sequence data.

As a first step, we consider modeling symmetry (sub)groups acting globally on the input; a natural next step is to incorporate local symmetries, which play a key role for molecular graph applications [Thiede et al., 2021; Zhang et al., 2024]). Another interesting direction is allowing the symmetry breaking object to be input-dependent. Incorporating "soft" equivariance priors in ASEN is another fruitful approach. Studying the scaling behavior of ASEN and the effect of symmetry model misspecification are key avenues for future work. Applying our framework for multi-physics pretraining [McCabe et al., 2024; Rahman et al., 2024] to model task-specific symmetry breaking is a promising direction. Finally, exploring other flexible forms of symmetry breaking and exploiting beyond permutation subgroups offer a promising path towards equivariant foundation models.

ACKNOWLEDGMENTS

The authors thank Erik Thiede (Cornell), Soledad Villar (JHU), Bruno Ribeiro (Purdue), Risi Kondor (UChicago), and Jinwoo Kim (KAIST) for valuable discussions. The authors also thank the anonymous reviewers and area chair for their constructive feedback. H.L. is supported by the Fannie and John Hertz Foundation. S.J. is supported by an Alexander von Humboldt professorship. This research was also partially supported by NSF AI Insitute TILOS.

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

# A    ADDITIONAL DETAILS OF ALGORITHM 1

1. **Lift the generators.** Given the generators $\sigma_1, \ldots, \sigma_r$ of $G \leq S_n$, we define new permutations $\rho_i$ acting on $X \times X$ with $X = \{1, \ldots, n\}$. Each $\rho_i$ acts diagonally:

$$\rho_i \colon (x_a, x_b) \;\mapsto\; (\sigma_i(x_a), \sigma_i(x_b)).$$

   In practice (e.g. in `sympy`), if $n$ is the degree of the action, we encode $(x_a, x_b)$ as a single index $a \cdot n + b$, and construct $\rho_i$ as a `Permutation` of size $n^2$.

2. **Form the diagonal subgroup.** Define

$$\Delta(G) \;=\; \langle \rho_1, \ldots, \rho_r \rangle,$$

   the subgroup of $S_{n^2}$ generated by the lifted permutations.

   Conceptually, we take the subgroup generated by the $\rho_i$. In `sympy`, this is done by calling

   ```
   Delta = PermutationGroup([ρ₁,...,ρ_r]).
   ```

   Internally, `PermutationGroup` builds a *base and strong generating set* (BSGS) for $\Delta(G)$ via Schreier–Sims. The BSGS consists of a chosen base $B = (b_1, \ldots, b_k)$ and *strong generators* adapted to the chain of stabilizers

$$\Delta(G) = G^{(0)} \geq G^{(1)} \geq \cdots \geq G^{(k)} = \{e\},$$

   where $G^{(i)}$ is the subgroup fixing the first $i$ base points in $B$. The associated *basic orbits* and *Schreier vectors* allow efficient navigation in the group.

3. **Compute the orbits.** The $G$–orbits on $X \times X$ are exactly the orbits of $\Delta(G)$. Concretely, starting from a pair $(x_a, x_b)$, we apply the generators $\rho_i$ repeatedly until no new pairs are found; the set obtained is its orbit. With a BSGS, orbit computation is polynomial-time as SymPy performs a breadth-first search on the strong generators and stores transversal information for reconstruction. In code, this is as simple as

   ```
   Delta.orbits()
   ```

   which returns all $\Delta(G)$-orbits of the action.

## B    PROOFS

**Proposition 1.**    *Let $h_\theta : \mathcal{X} \times \mathcal{V} \to \mathcal{Y}$ be **G**-equivariant, and let $\mathrm{Aut}(\mathbf{v}) = G$. Then $f_\theta(x) := h_\theta(x, \mathbf{v})$ is equivariant to $G$. If additionally $h_\theta$ is injective in the input $\mathbf{v}$, then $f_\theta$ is not equivariant to any transformation in $\mathbf{G} \setminus G$.*

*Proof.* We have already shown in Section 3.1 that $f_\theta$ is $G$-equivariant. Now, suppose that $h_\theta$ is injective in $\mathbf{v}$, and let $g \in \mathbf{G} \setminus G$. We want to show that $f_\theta(gx) \neq gf_\theta(x)$ for some choice of $x$. We choose any $x \in \mathcal{X}$. This holds because $\mathbf{v} \neq g\mathbf{v}$ (since otherwise $g \notin G = \mathrm{Aut}(\mathbf{v})$). Thus, by injectivity, $h_\theta(gx, \mathbf{v}) \neq h_\theta(gx, g\mathbf{v})$. This allows us to conclude that

$$f_\theta(gx) = h_\theta(gx, \mathbf{v}) \tag{10}$$
$$\neq h_\theta(gx, g\mathbf{v}) \tag{11}$$
$$= gh_\theta(x, \mathbf{v}) \tag{12}$$
$$= gf_\theta(x). \tag{13}$$

This concludes the proof.    $\square$

We remark that the injectivity of $h_\theta$ in the input $\mathbf{v}$ is sufficient to prove Prop. 1, but not necessary. As shown in the proof, this injectivity assumption allows us to show $f_\theta(gx) \neq gf_\theta(x)$ for *any* $x \in \mathcal{X}$, while establishing this for a particular choice of $x$ suffices.

**Lemma 1.**    *If a one-layer MPNN $h_\theta$ uses injective functions for (hyper-)edge feature update $\psi_e$, node update $\phi$, and (hyper-)edge multiset aggregation $\tau$, and if the node features are distinct, then $h_\theta$ is not equivariant to permutations in $S_n \setminus G$ where $G = \mathrm{Aut}(A^{(2)})$.*

*Proof.* We will show that for any $\sigma \in S_n \setminus G$, $f_\theta(\sigma X) \neq \sigma f_\theta(X)$. By (9), it suffices to show that there exists a node $i$ such that

$$h_\theta(\sigma X, A^{(2)})[i] \neq h_\theta(\sigma X, \sigma A^{(2)})[i]. \tag{14}$$

Since $\sigma \in S_n \setminus G$, there exists a node pair $(i, j)$ such that $A^{(2)}_{i,j} \neq A^{(2)}_{\sigma(i),\sigma(j)}$. By the assumption that the node features $\{X_j\}_{j=1}^n$ has distinct elements, the multisets

$$\{\!\!\{ (X_\sigma(i), X_\sigma(j), A^{(2)}_{i,j}) \mid j \in [n] \}\!\!\} \neq \{\!\!\{ (X_\sigma(i), X_\sigma(j), A^{(2)}_{\sigma(i),\sigma(j)}) \mid j \in [n] \}\!\!\}. \tag{15}$$

By definition of MPNN (9),

$$h_\theta(\sigma X, A^{(2)})[i] = \phi\left( \psi_n(X_\sigma(i)), \tau\left( \{\!\!\{ \psi_e(X_\sigma(i), X_\sigma(j), A^{(2)}_{i,j}) \mid j \in [n] \}\!\!\} \right) \right) \tag{16}$$

$$h_\theta(\sigma X, \sigma A^{(2)})[i] = \phi\left( \psi_n(X_\sigma(i)), \tau\left( \{\!\!\{ \psi_e(X_\sigma(i), X_\sigma(j), A^{(2)}_{\sigma(i),\sigma(j)}) \mid j \in [n] \}\!\!\} \right) \right). \tag{17}$$

By injectivity of $\psi_e$ and (15),

$$\{\!\!\{ \psi_e(X_\sigma(i), X_\sigma(j), A^{(2)}_{i,j}) \mid j \in [n] \}\!\!\} \neq \{\!\!\{ \psi_e(X_\sigma(i), X_\sigma(j), A^{(2)}_{\sigma(i),\sigma(j)}) \mid j \in [n] \}\!\!\}.$$

By injectivity of $\tau$ and $\phi$, we have $h_\theta(\sigma X, A^{(2)})[i] \neq h_\theta(\sigma X, \sigma A^{(2)})[i]$, completing the proof.    $\square$

**Theorem 1.**    *Any order 1 G-MLP can be approximated to arbitrary accuracy on a compact domain via ASEN with $K = 2$ and an MPNN base model $h_\theta$.*

*Proof.* We show that one layer of ASEN using a message-passing GNN backbone can simulate $\sigma \circ T_i$, so suppose $L = 1$ (i.e. the G-MLP has one layer). Recall that any equivariant linear map $T : \mathbb{R}^n \to \mathbb{R}^n$ can be viewed as a linear combination $T = \sum_{l=1}^d a_l B^l$, where $a_l \in \mathbb{R}$ are scalars, the $B^l : \mathbb{R}^n \to \mathbb{R}^n$ are G-equivariant linear maps that span the vector space of G-equivariant linear maps, and $d$ is the dimension of this vector space of G-equivariant linear maps. Moreover, by an argument similar to Maron et al. [2019], we can define the $B^l$ as follows:

Let $\tau_1, \ldots, \tau_q$ be the unique orbits of the action of $G$ on node-pair indices $[n] \times [n]$ (we refer to these as node-pair orbits). Then Ravanbakhsh et al. [2017]; Maron et al. [2019] show that $q = d$, and the $B^l$ can be chosen as:

$$B_{ij}^l = \begin{cases} 1 & (i,j) \in \tau_l \\ 0 & \text{else.} \end{cases} \tag{18}$$

This is a form of weight sharing, where $B^l$ is constant on each node-pair orbit (and hence any $T$ that is a linear combination of them is constant on each node-pair orbit). The index $l$ denotes the $l$-th node-pair orbit. Note that the linear map can be shown to take a message-passing form as follows. For an input $x \in \mathbb{R}^n$, let $x_i$ be viewed as a node representation for node $i$. Then the new node representation for node $i$ after this layer is

$$T(x)_i = \sum_{l=1}^q a_l (B^l x)_i \tag{19}$$

$$= \sum_{l=1}^q a_l \sum_{j=1}^n B_{ij}^l x_j \tag{20}$$

$$= \sum_{j=1}^n x_j \sum_{l=1}^q a_l B_{ij}^l \tag{21}$$

This can be interpreted as message passing, where the node $j$ passes message $x_j \sum_{l=1}^q a_l B_{ij}^l$ to the node $i$. We will show that ASEN with order $K = 2$ and a message-passing GNN $h_\theta$ can simulate this map $T(x)$. Note that any $H \in \mathbb{R}^{n^2}$ that has $\text{Aut}(H) = G^{(2)}$ must satisfy that $H_{i_1, j_1} = H_{i_2, j_2}$ if and only if $(i_1, j_1) \sim_G (i_2, j_2)$. In other words, $H_{i_1, j_1} = H_{i_2, j_2}$ if and only if $(i_1, j_1)$ and $(i_2, j_2)$ are in the orbit $\tau_l$ for some $l$. Let the distinct entries of $H$ be denoted by $h_l \in \mathbb{R}$, so that $h_l = H_{i_1, j_1}$ if and only if $(i_1, j_1) \in \tau_l$.

Finally, we define the GNN $h_\theta$ to take the following message-passing-based form. Let $x_i$ be the representation of node $i$. The GNN updates the node representations to $\hat{x}_i$ via two multilayer perceptrons $\text{MLP}^e(h) : \mathbb{R} \to \mathbb{R}^q$ and $\text{MLP}^v : \mathbb{R}^{q+1} \to \mathbb{R}$ as follows:

$$\hat{x}_i = \sum_{j=1}^n \text{MLP}^v(x_j, \text{MLP}^e(H_{i,j})) \tag{22}$$

$$\text{MLP}^e(h)_l = \begin{cases} 1 & \text{if } h = h_l \\ 0 & \text{else} \end{cases} \tag{23}$$

$$\text{MLP}^v(x, y) = x \sum_{l=1}^q a_l y_l. \tag{24}$$

Note that $\text{MLP}^e(H_{i,j}) = B_{ij}^l$ so that $\text{MLP}^v(x_j, \text{MLP}^e(H_{i,j})) = x_j \sum_{l=1}^q a_l B_{ij}^l$, which shows that $\hat{x}_i = T(x)_i$. In practice, an MLP cannot exactly express these functions, but an MLP can approximate each function to arbitrary precision $\epsilon > 0$ on a compact domain. Note that the function $\text{MLP}^e$ seems discontinuous, but it is only defined on finitely many inputs, so it has a continuous extension that is exact on the finite inputs. □

**Theorem 2.** *Let* $\mathbf{G}$ *be a compact group,* $\mathcal{X}, \mathcal{V}$ *be compact metric* $\mathbf{G}$-*spaces, and* $\mathcal{Y}$ *be a compact* $\mathbf{G}$-*space. Let* $f_\theta : \mathcal{X} \times \mathcal{V} \to \mathcal{Y}$ *be a universal family of continuous* $\mathbf{G}$-*equivariant networks, i.e.* $f_\theta(gx, gv) = g \cdot f_\theta(x, v)$. *Consider* $\mathcal{H} \in \mathcal{V}$ *with stabilizer equal to a subgroup* $G \le \mathbf{G}$. *Then, the family* $f_\theta(\cdot, \mathcal{H})$ *is universal over continuous* $\mathbf{G}$-*equivariant functions from* $\mathcal{X}$ *to* $\mathcal{Y}$.

*Proof.* Let $f^* : \mathcal{X} \to \mathcal{Y}$ be a continuous, $G$-equivariant function. To prove universality of $f_\theta(\cdot, \mathcal{H})$, we must show that for any $\epsilon$, there exists a $\theta$ such that $f_\theta(\cdot, \mathcal{H})$ is $\epsilon$-close to $f^*$. To achieve this, let's first define a new function (which we will prove is $\mathbf{G}$-equivariant), $n : \mathcal{X} \times \mathcal{O} \to \mathcal{Y}$ where $\mathcal{O} = \{g\mathcal{H} : g \in \mathbf{G}\}$ as follows: for any $g \in \mathbf{G}$,

$$n(x, g\mathcal{H}) := g f^*(g^{-1} x). \tag{25}$$

Note in particular that $n(x, \mathcal{H}) = f^*(x)$. To first show that is a valid definition of $n$, we will argue that if $g\mathcal{H} = g'\mathcal{H}$, then $gf^*(g^{-1}x) = g'f^*(g'^{-1}x)$ for all $x \in \mathcal{X}$. By the assumption $g\mathcal{H} = g'\mathcal{H}$, we have $g' = gs$ for some $s \in G$, where $G$ is the stabilizer of $\mathcal{H}$. Then

$$g'f^*(g'^{-1}x) = (gs)f^*((gs)^{-1}x) = g(sf^*(s^{-1}g^{-1}x)) = gf^*(g^{-1}x), \qquad (26)$$

where the last equality follows from $f^*$ being $G$-equivariant.

Next, the continuity of $n$ on $\mathcal{X} \times \mathcal{O}$ follows from the continuity of $f^*$, the inversion map $g^{-1}$ and group action $g$.

Finally, to show $n$ is $\mathbf{G}$-equivariant, we note that for any $h \in \mathbf{G}$,

$$n(hx, hg\mathcal{H}) = hgf^*\left((hg)^{-1}hx\right) = hgf^*(g^{-1}x) = hn(x, g\mathcal{H}). \qquad (27)$$

Thus, $n$ is a continuous $\mathbf{G}$-equivariant function on $\mathcal{X} \times \mathcal{O}$.

By universality of $\{f_\theta\}$ on $\mathcal{X} \times \mathcal{V} \mapsto \mathcal{Y}$ (under the supremum norm) and the orbit $\mathcal{O}$ being closed in $\mathcal{V}$, $\{f_\theta\}$ is also universal on the subdomain $\mathcal{X} \times \mathcal{O}$. In particular, for any $\epsilon$, there exists some $\theta$ such that $f_\theta$ approximates $n$ $\epsilon$-well over all of $\mathcal{X} \times \mathcal{O}$. Thus, it must also approximate $n$ $\epsilon$-well over $\mathcal{X} \times \{\mathcal{H}\}$. But since $n = f^*$ on $\mathcal{X} \times \{\mathcal{H}\}$, this completes the proof.

$\square$

We remark that Thm. 2 can be generalized to locally compact groups, via Jaworowski's equivariant extension theorem [Jaworowski, 1976; Lashof, 1981].

## C  ADDITIONAL EXPERIMENT DETAILS

### C.1  DETAILS FOR PATHFINDER TASK

The Pathfinder task from Long Range Arena [Tay et al., 2021] is a binary image classification problem: determine if there exists a path connecting two marked pixels. The image is flattened as a sequence input to probe the spatial reasoning of sequence models (e.g. Transformers). It is known in the literature that standard Transformers perform poorly on Pathfinder with 1D positional encoding (unless with proper pretraining [Amos et al., 2024]), whereas 2D positional encoding (PE) provides better inductive bias by mapping the pixel $(i, j)$ as $E_R[i] + E_C[j]$ where $E_R, E_C \in \mathbb{R}^{m \times d}$ are row and column embeddings (unlike 1D-PE embedding $E \in \mathbb{R}^{m^2 \times d}$ for the flattened grid of size $m \times m$).

We use a standard Transformer architecture for Pathfinder task taken from `https://github.com/mlpen/Nystromformer`. We use the default model configuration (e.g., 2 layers, 2 heads, 64 embedding dimensions, learnable positional encoding). We follow the standard protocol of training the model for 62400 steps, linear learning rate decay schedule, and perform hyper-parameter search over the learning rate $\{0.0005, 0.0001, 0.00008\}$. We report more details of the performance for 1D-PE and 2D-PE variants in Tab. 4.

Table 4: Performance of Transformers in `Pathfinder64` tasks, using different positional encoding (PE): 1D-PE (learnable), 2D-PE (learnable, separate for rows and columns) with different group $G$. The accuracy is averaged over 3 runs.

| Method | Group $G$ | Params. | Test Acc. (mean $\pm$ std) | Train Acc. (mean $\pm$ std) |
|---|---|---|---|---|
| 1D-PE | $I$ | 370.5k | $0.656 \pm 0.135$ | $0.816 \pm 0.093$ |
| 2D-PE | $I$ | 116.6k | $0.818 \pm 0.005$ | $0.831 \pm 0.006$ |
| | $(S_4)^{1024}$ | 112.5k | $0.824 \pm 0.000$ | $0.848 \pm 0.006$ |
| | $(S_9)^{455}$ | 111.2k | $0.827 \pm 0.018$ | $0.852 \pm 0.013$ |
| | $(S_{16})^{256}$ | 110.4k | $0.814 \pm 0.020$ | $0.840 \pm 0.009$ |

### C.2  DETAILS FOR SYNTHETIC SEQUENCE MODELING TASKS IN SEC. 5.2

We describe in details our chosen synthetic tasks in Tab. 5.

**Equivariant Experiment Set-up**  The dataset per task contains 2500 examples of sequence length 10 and vocabulary size 7, chosen to highlight the benefits of symmetry in data-scarce regimes. We train ASEN in the binary node classification setting to learn an equivariant mapping from the input node features to output node labels, both input and output represented as sequences. Our model is optimized using standard cross-entropy loss, with hidden dimension of 128, batch size of 64, learning rate of 0.01, and run on 40 epochs.

**Invariant Experiment Set-up and Results**  We apply ASEN in the invariant setting on the following synthetic tasks: Palindrome, Intersect, Detect Capital and Longest Palindrome (c.f. Tab. 3 and Tab. 5). As each task can have a different type of label, we switch to the regression setting using an $L_1$ loss and train for 80 epochs. The experiment uses a data-scarce regime where each task has 8000 training datapoints, denoted as 1 unit. As shown in Tab. 6, ASEN with the correct symmetry group outperforms its trivial symmetry counterpart across tasks and training dataset sizes $r \in \{1, 1.5, 2\}$. We further perform an ablation study on the effect of training set size. As shown in Fig. 9, for low-data regimes ($r \leq 1.5$), all tasks benefit from imposing invariance; with more training data ($r \geq 2$), some tasks (e.g., Detect Capital) exhibit comparable performance between invariant ASEN and its non-invariant counterpart.

### C.3  DETAILS FOR MULTITASK APPLICATIONS SEC. 5.2.1

In Fig. 6, we conduct multitask training of ASEN across Intersect, Cyclicsum, and Palindrome, and observe that the performance on Intersect notably improved upon single task training. Meanwhile, Tab. 7 shows that the multitask performance on the other two tasks (Cyclicsum, Palindrome) remains

Table 5: Description of synthetic tasks, equivariant and invariant learning set-up, and their corresponding symmetry group.

| | Synthetic Tasks | | |
|---|---|---|---|
| Task | Equivariant | Invariant | Symmetry |
| Intersect | Given two sequences of length $\frac{n}{2}$, determine which elements of each sequence are present in the other | Determine the size of the intersection of the two sequences | $S_{n/2} \times S_{n/2} \times S_2$: sequences can be reordered, and the two sequences can be swapped |
| Symmetric Difference | Given two sequences of length $\frac{n}{2}$, determine which elements of each sequence are *not* present in the other | Determine the size of the difference of the two sequences | |
| Palindrome | Given a sequence of length $n$, determine where the sequence has a contiguous subsequence that is a palindrome of length $k$, if one exists | Determine if the sequence has a contiguous subsequence that is a palindrome of length $k$ | Sequence reversal |
| Monotone Run | Given a sequence of length $n$, determine where the sequence has a contiguous subsequence that is strictly increasing or decreasing of length $k$, if one exists | Determine if the sequence has a contiguous subsequence that is strictly increasing or decreasing of length $k$ | |
| Cyclic Sum | Given a sequence of length $n$, determine which cyclic contiguous subsequence of length $k$ has the largest sum | Find the largest sum of a cyclic contiguous subsequence of length $k$ | $C_n$ (cyclic shifts) |
| Cyclic Product | Given a sequence of length $n$, determine which cyclic contiguous subsequence of length $k$ has the largest product | find the largest product of a cyclic contiguous subsequence of length $k$ | |
| Detect Capital | N/A | Given a string of length $n$, return True if properly capitalized, meaning the string is all Uppercase, lowercase, or only has first letter capitalized | $S_{n-1}$, as all elements except first can be permuted |
| Longest Palindrome | N/A | Given a string of length $n$, determine the length of the longest palindrome that can be constructed using the characters of the string | $S_n$ |

Table 6: Performance of two ASEN variants across tasks for varying data amounts. Within each model variant, columns indicate data used (in units): 1, 1.5, and 2.

| ASEN (Loss ↓) | Invariant | | | Non-invariant | | |
|---|---|---|---|---|---|---|
| Task | 1 | 1.5 | 2 | 1 | 1.5 | 2 |
| Intersect | $0.076 \pm 0.002$ | $0.061 \pm 0.009$ | $0.060 \pm 0.005$ | $0.081 \pm 0.002$ | $0.078 \pm 0.0004$ | $0.077 \pm 0.001$ |
| Palindrome | $0.250 \pm 0.04$ | $0.170 \pm 0.01$ | $0.125 \pm 0.05$ | $0.270 \pm 0.04$ | $0.250 \pm 0.04$ | $0.132 \pm 0.04$ |
| Detect Capital | $0.053 \pm 0.005$ | $0.056 \pm 0.007$ | $0.068 \pm 0.016$ | $0.075 \pm 0.011$ | $0.066 \pm 0.004$ | $0.063 \pm 0.007$ |
| Longest Palindrome | $0.138 \pm 0.014$ | $0.110 \pm 0.009$ | $0.110 \pm 0.009$ | $0.152 \pm 0.007$ | $0.130 \pm 0.012$ | $0.130 \pm 0.010$ |

similar to the single task setting. Figure 10 shows the compute cost with respect to the training set size and the number of tasks, which grows linearly with both factors.

Table 7: Multi-task Test Losses (0.08 units)

| Method | Cyclicsum | Palindrome |
|---|---|---|
| Single-task | 0.3869 | 0.510 |
| Multi-task | 0.3839 | 0.537 |

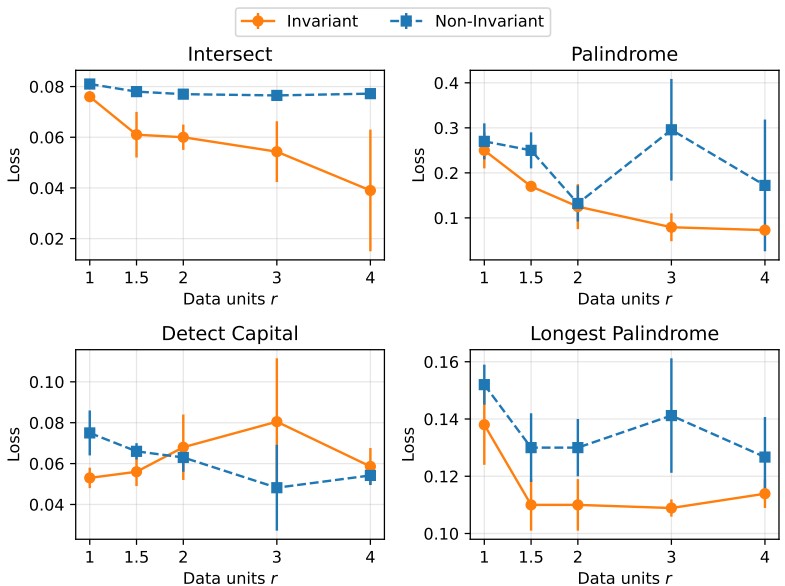

Figure 9: Invariant experiment ablation: test loss of ASEN using invariance (orange) versus its non-invariant counterpart (blue) across varying training data sizes (1 unit refers to 8,000 datapoints) and different tasks.

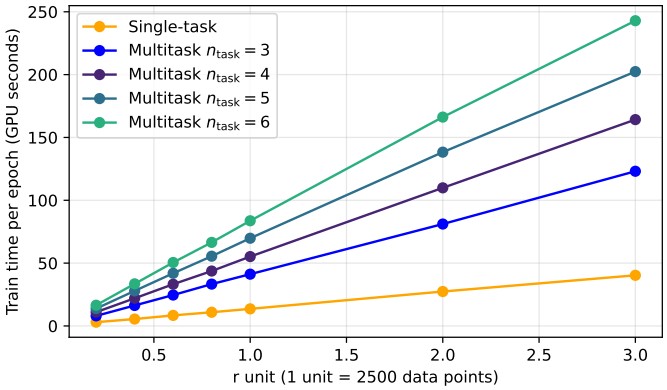

Figure 10: Compute cost in multitask training (measured as training time per epoch on an NVIDIA A100 GPU device) with respect to the number of training set sizes $r$ and the number of tasks $n_{\text{task}}$.

## C.4 DETAILS FOR TRANSFER LEARNING SEC. 5.2.2

For the invariant setting, we adopt a weighted $L_1$ regression loss for training (restricting all losses to between 0 and 1), and use standard $L_1$ loss for evaluation. We disable the TokenEmbedder, due to the diverse (invariant) target range across tasks. The tasks Palindrome, Intersect, Detect Capital and Longest Palindrome are each learned on an instance of ASEN pretrained on the other three. Each task dataset is of size 5600 datapoints, with the chosen (finetuned) dataset restricted to 15 percent, or 840 datapoints. To probe the effect of transferring symmetry, we also provide the ASEN baseline with the trivial symmetry (denoted as "NoSym"), which underperforms ASEN with the desired nontrivial symmetry group in the data-scarce regime.

We also ablate how the pretraining effect varies with input sequence length. We use the same tasks and set-up as the transfer learning experiment described above, except using a smaller task dataset size with 4000 datapoints and varying the input sequence length over $\{6, 12, 18\}$. As shown in Fig. 11, pretraining effect depends on the task and the sequence length. Increasing the sequence length leads

to a larger symmetry group, but also introduces greater inter-group difference across tasks. Identifying optimal task mixture and designing more effective training curriculum is an interesting direction for future work.

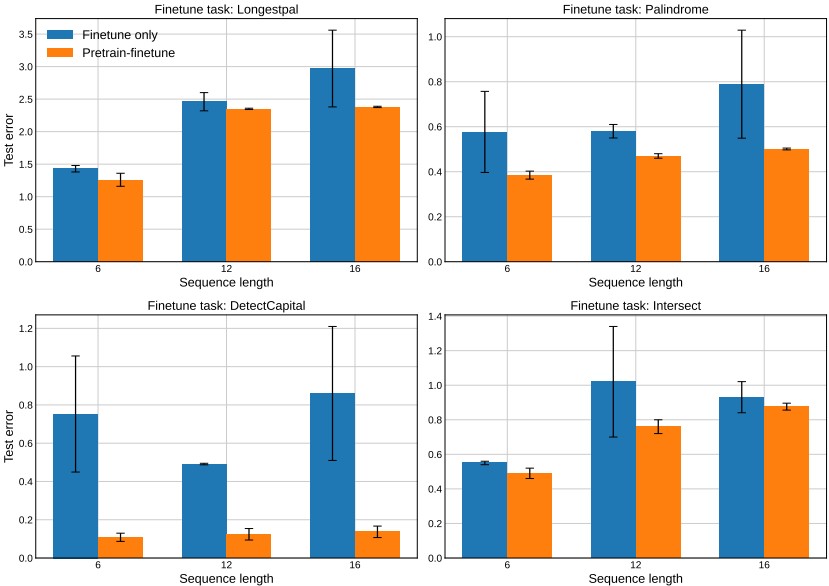

Figure 11: Invariant pretraining ablation: test error of ASEN for finetune-only (blue bars) versus pretrain-finetune (orange bars) across varying input sequence lengths. Each subplot shows the pretraining effect for a different finetune task.

## C.5 OVERCONSTRAINED SYMMETRY

In Sec. 5.2 and Sec. 5.2.2, we show that ASEN incorporating the desired symmetries outperforms its trivial symmetry baseline across a wide range of tasks and settings. As noted before, the symmetry groups considered in these tasks are equal to their 2-closure, and thus ASEN can accurately model the desired symmetry via the edge orbits whose automorphism are the 2-closure group. We also discuss in Sec. 5 that ASEN can learn more symmetries from data (i.e., learn to tie weights in the *EdgeEmbedder* module), making it robust when choosing $G^{(2)}$ to be smaller than the target symmetry. We now present a negative example where $G^{(2)} > G$; in such a case, ASEN can fail.

Consider the task of determining the sign of a permutation, namely deciding whether an even or odd number of inversions is used to create a permutation. For example, for $n = 4$, the identity permutation $[1, 2, 3, 4]$ is even, whereas $[2, 1, 3, 4]$ with one inversion is odd, and $[2, 3, 1, 4]$ with two inversions is even. This task is invariant under the alternating group $A_n$, while the 2-closure of $A_n$ is $S_n$, meaning using ASEN taking the 2-closure symmetry breaking input would include significantly more symmetries.

Choosing permutations of length $n = 7$, we train ASEN with $G^{(2)} = S_n$ and its trivial symmetry baseline using binary cross-entropy loss for this invariant classification problem. We find that ASEN performs no better than chance (i.e., stuck at test loss of $0.69$, the same as a random guess baseline). On the other hand, the trivial symmetry baseline eventually reaches a test loss of $0.28$, outperforming ASEN with overconstrained symmetries.

