# OpenReview forum: "Any-Subgroup Equivariant Networks via Symmetry Breaking"
_ICLR.cc/2026/Conference — ICLR 2026 Poster_

### Official Review · Reviewer_VJcc · 2025-10-21

**Soundness:** 3
**Presentation:** 4
**Contribution:** 2
**Rating:** 8
**Confidence:** 3

**Summary:**

The paper proposes using networks equivariant under a group H, together with a symmetry breaking input to obtain equivariance under a subgroup G of H. Specifically, H is taken to be the symmetric group on n elements, where n is the dimensionality of the input. Symmetry breaking is obtained by designing edge features that define a graph that has automorphism group $G^{(2)}$, the 2-closure of G. $G^{(2)}$-equivariance is obtained by using a graph network on the constructed colored graph. When G is 2-closed, this yields equivariance precisely under G.

**Strengths:**

I found the paper well written and engaging. The proposed method is well thought out and deserves to be discussed at the conference in my opinion. Theorem 1 was especially interesting to me, showing that equivariant MLPs can be simulated by the proposed graph networks.

**Weaknesses:**

The experimental evaluation seems not so convincing to me:

1. There are no comparisons to equivariant MLPs.
2. In Figure 6 it looks like just training the yellow models longer will allow them to catch up. Is the effective training time here the same for the blue and yellow models or do the blue models train for longer due to training on multiple tasks?
3. There is an example in the appendix where $G$ is not 2-closed, and then the method does not work.


**Typos:**

I believe that the symmetry group of the intersection operation in table 3 should not be a direct product since permuting the two sets does not commute with permuting internally in one set.

**Questions:**

1. I am not very familiar with the prior work “Approximately Equivariant Graph Networks” by Huang et al. Is it correct to state that they propose using ordinary MLPs equivariant under the automorphism group of a fixed graph to obtain more expressive networks for that specific graph, whereas the submitted paper does the opposite of proposing to design a graph with a specific automorphism group G to obtain G-equivariant networks through a graph network?
2. In table 1, the lowest error is consistently obtained by the non-equivariant version, which, if I understand correctly, corresponds to a unique identifier for each edge. Is this a standard approach in graph networks on fixed graphs?

As I found the paper interesting I have some potentially more philosophical questions.

3. As far as I understand, the suggested approach requires knowing the orbits of the node pairs under G, but from this information we can also construct a basis for the equivariant linear maps (Ravanbakhsh et al, 2017). So is the following statement in the introduction fair? “(I) equivariant architectures typically require deriving and implementing group-specific equivariant layers, so substantial research and engineering must be done for architectural design whenever a new type of symmetry arises,”
4. A linear $G$ equivariant layer from $\mathbb{R}^n$ to $\mathbb{R}^n$ under the permutation action $\rho(g)$ can be defined from a general linear map $W\in \mathbb{R}^{n\times n}$ through group averaging as $W_\rho = \sum_{g\in G}\rho(g) W \rho(g)^{-1}$. Thus the same $W$ can define equivariant linear layers for different groups $G$ acting on $\mathbb{R}^n$. Is this somehow analogous to how different symmetry breaking objects lead to equivariance under different $G$ for the same message passing layer in the paper?
5. In some sense, the paper suggests an approach that enables the use of graph networks instead of equivariant MLPs. Aren't equivariant MLPs easier to work with, and more aligned with current hardware, due to the focus on linear layers?

---

> ### Author Response · Authors · 2025-11-24
>
> We thank the reviewer for their encouraging feedback, in particular their appreciation of the novelty and clarity of our paper. We provide detailed responses to concerns raised in the Weakness and Questions section.
>
> ---
>
> **W1 Comparison to equivariant MLP**
> * Single-task setting. For the graph tasks, the baseline method from Huang et al. essentially constructs an equivariant MLP: it interleaves $G$-equivariant linear layers with pointwise nonlinearity, where a $G-equivariant linear layer is obtained by weight sharing (similar to Ravanbakhsh et al, 2017). From Sec 5.1.1, we see that our framework ASEN achieves similar performance as the equivariant MLP from Huang et al..
> * Multitask settings (Sec 5.2.1). In contrast, for multitask applications, equivariant MLP is similar to the single-task baseline (Fig.6). This is due to the weights of equivariant MLP being specific to the particular chosen group (task), and not transferable across tasks. In contrast, our method ASEN uses the shared model weights across groups (tasks), requiring only a change of the symmetry breaking input (edge features) for each task.
> * Connection to group averaging. We further discuss the group-averaging method to obtain equivariant linear layers and its connection to our method in our response to Q3 below.
>
> **W2 Training time comparison**
>
> We thank the reviewer for raising this point. We have included the training time cost with respect to the number of tasks and the training set size in Figure 10. As the reviewer correctly pointed out, the multitask setting (blue curve) incurs a higher training cost—approximately a factor of the number of tasks compared to the single-task baseline. To make a fairer comparison, we have now added a dashed reference line in Figure 6 (top), showing the multitask performance at epoch 13, which matches the total training time of the single-task baseline at epoch 40. Even under this matched-time comparison, we see that for low-data regimes ($r \leq 0.6$), there is still a notable performance gain from multitask training.
>
> **W3 Limitation of ASEN when the 2-closure group significantly differs from the group**
>
> We agree with the reviewer and acknowledge this as a limitation of our method. However, we would like to point out that there are many groups where their 2-closures are equal to the group. These include ﬁnite nilpotent groups that are either cyclic or a direct product of a generalized quaternion group with a cyclic group of odd order [1], and all the groups used in our experiments from the main paper. We have added more discussion on the 2-closure approximation in the revised paper (line 177-182).
>
> **Q1 Comparison with Huang et al.**
>
> Yes – that’s a great summary contrasting our approach with that of Huang et al.
>
> **Q2 Table 1 Lowest Error from non-equivariant baseline**
> * Your interpretation is correct. However, this unique-edge-orbit approach for fixed graphs is not standard in graph networks. Prior to Huang et al., the default message-passing backbone typically uses two edge orbits, one for self-edges, and one for all remaining edges. This effectively models the full permutation symmetry, which can be too restrictive for learning on the fixed graph setting, as shown theoretically and empirically in Huang et al. In contrast, using the unique-edge orbit corresponds to modelling the trivial symmetry group.
> * The improved performance under the trivial symmetry group is likely due to the specifics of this application: the human pose graph is rather small (16 nodes), so maximizing expressivity with enough training data yields the best results.
>
> **Q3 Introduction statement**
>
> We thank the reviewer for clarifying this statement. Although we can use the edge orbits to construct equivariant MLPs, a new equivariant MLP layer needs to be implemented when a different symmetry group arises. Different $G$-equivariant MLP layers admit different linear layer structure (i.e., different weight sharing pattern in the weight matrix), which make them hard to transfer across different group $G$. If the reviewer feels strongly about this, we are happy to expand the sentence for clarifications or tone it down suitably.

---

> ### Author Response · Authors · 2025-11-24
>
> **Q4 Connection between group averaging and our symmetry breaking method**
>
> We thank the reviewer for raising this excellent point. Indeed they are related by interpreting the group averaging operator from the perspective of the node orbits. Concretely:
> Let $(i,j) \sim (i’, j’) \in \mathcal{O}$ be in the same node pair orbit.
> Then the group-averaged equivariant linear map has its $(i,j)$-entry given by
>
> $$\begin{align}
> \bar{W}[i,j] = \frac{1}{|G|} \sum_g \rho(g) W[i,j] \rho(g)^{-1} = \frac{1}{|G|} \sum_g W[g^{-1}(i), g^{-1}(j)].
> \end{align}$$
>
> Grouping the sum by orbit elements, we have
>
> $$
> \bar{W}[i,j]
> = \frac{1}{|G|} \sum_{(i’,j’) \in \mathcal{O}} |\text{stab}(i,j)| W[i’, j’] =\frac{1}{|\mathcal{O}|}  \sum_{(i’,j’) \in \mathcal{O}} W[i’, j’],
> $$
> where the last equality follows from the orbit stabilizer theorem. This shows that the group averaged $G$-equivariant linear map $\bar{W}$ has entries being constant over node pair orbits. This is precisely the form $B_{i,j}^{l}$ in equation (17). The connection to message-passing and our symmetry-breaking edge features follow from the rest of the proof in Theorem 1.
>
> **Q5 Comparison of graph networks with equivariant MLP**
>
> * *Computational cost*. Both equivariant MLP and graph networks have similar costs ($O(n^2)$ for dense weight matrices and graphs, and $O(|E|)$ with sparsity constraints).
> * *Flexibility of our approach*. The advantage of our proposed graph networks lies in its flexibility: different symmetries can be implemented simply by modifying the symmetry-breaking edge features. In contrast, equivariant MLP typically requires a new equivariant layer design for each different group, which demands more engineering resources.
> * *Cost during training*. The group averaging method discussed in Q4 can turn arbitrary linear maps into suitable $G$-equivariant maps, but doing so incurs an $O(n^2)$ cost *during training*. On the other hand, our framework only requires a one-time preprocessing step to construct edge features, and does not incur additional costs during training.
>
> ---
>
> We thank the reviewer for helping improve the quality of our paper, and remain available to address any further questions or concerns.
>
> References
>
> 1. Abdollahi and Arezoomand, Finite nilpotent groups that coincide with their 2-closures in all of their faithful permutation representations, J. Algebra Appl. 17(4)(2018) 1850065.

---

> > ### Comment · Reviewer_VJcc · 2025-11-27
> >
> > Thank you for the clear answers to my and other reviews. I believe that my initial positive score is fair and will keep it.
> >
> > To my understanding, implementing equivariant MLPs in the single task case is not more difficult than constructing a symmetry breaking object as required in ASEN (indeed, the prior work by Huang et al does construct such MLPs). So the killer application for ASEN would be the multitask setting, but the experiments in that setting are only on small synthetic sequence processing tasks.

---

> > > ### Author Response · Authors · 2025-12-01
> > >
> > > Thank you again for engaging with our rebuttal, and for recommending our paper for acceptance. We agree with the reviewer that the killer application of ASEN lies in the multitask (and transfer learning) setting. Our experiments focus on synthetic tasks to provide a clean proof of concept for this advantage. We further conjecture that applying ASEN to other large-scale applications, such as multiphysics pretraining (see details of the response to 8QJx), could yield a more expressive and transferable foundation model; we plan to explore this direction in future work. We will incorporate these discussions into the revised version of the paper.

---

### Official Review · Reviewer_gDLb · 2025-10-28

**Soundness:** 4
**Presentation:** 3
**Contribution:** 3
**Rating:** 8
**Confidence:** 3

**Summary:**

The paper proposes a simple framework to build neural networks that are equivariant to any subgroup of a given symmetry group. By conditioning the network on a symmetry-breaking input $v$ whose automorphism $Aut(v)$ equals the desired subgroup, the model naturally inherits the correct equivariance. The approach is general, easy to apply, and shown to work well on several examples involving rotation and reflection subgroups.

**Strengths:**

- Presents an interesting and elegant idea that is both simple and flexible for constructing subgroup-equivariant networks.

- Provides a solid mathematical foundation, with clear and rigorous proofs of the main results.

- Includes several illustrative examples that effectively clarify the concept and demonstrate its broad applicability.

- Offers a unifying framework that can reproduce or extend many existing equivariant architectures with minimal effort.

**Weaknesses:**

Overall, I find the paper is convincing and I am not aware of any similar work, therefore the weaknesses below should be regarded as minor:

- Computational efficiency and scalability are not analyzed, leaving open how well the approach performs for larger models or groups with respect to alternative approaches.

- The experiments demonstrate flexibility, but not whether this is the preferred method compared to specialized architectures; it may mainly serve as a prototyping tool.

- The experimental validation is limited and does not convincingly show advantages on more challenging or large-scale tasks.

**Questions:**

The framework offers flexibility to easily test different equivariances. Do the authors view it mainly as a prototyping system, or do they expect it to be competitive with specialized architectures in practical applications?

---

> ### Author Response · Authors · 2025-11-24
>
> We thank the reviewer for their encouraging feedback, in particular their appreciation of the *novelty and elegance* of our framework, and their acknowledgement of our technical and practical contributions. We provide detailed responses to the minor concerns raised in the Weakness and Questions section.
>
> ---
>
> **W1 Computational efficiency and scalability**
>
> We appreciate the reviewer raising this point, and we have added the computational costs of our framework (line 299-302), together with empirical training time costs (Figure 10). In particular, the preprocessing cost (algorithm 1) scales as $O(r n^2)$ where $n$ is the number of nodes and $r$ is the number of the generators of the group $G$. The training cost with a message-passing neural network backbone scales as $O(m n^2)$ on a fully-connected dense graph where $m$ is the number of training data points, and $O(m |E|)$ on a sparse graph where $|E|$ is the number of edges. This is comparable to alternative methods such as equivariant MLP (see more details in response to VJcc — Q5 Comparison of graph networks with equivariant MLP).
>
> **W2/Q1 Comparison of our framework versus specialized architectures**
>
> * Our motivation is to have a generalist model with a hint of specialization, addressing the growing tension between generalist and specialist models in modern ML. We show evidence that this works better than task-specific models using the same backbone (e.g. see multitask experiments in Sec. 5.2.1).
> * We interpret "specialized architectures" asked by the reviewer as domain-specific models that incorporate task-specific feature engineering, task-specific training curriculum, etc. Then without using this task-specific knowledge in our backbone, we do not expect our model to be as competitive.
> * That said, one could try to incorporate this task-specific knowledge in the preprocessing or post-processing pipeline of our framework, which could further improve its competitiveness.
> * Finally, we expect that it also depends on how similar the specific tasks are: with increased similarity, our generalist model can extract shared knowledge across tasks, and outperform a task-specific model (e.g., see pretraining experiments in Sec. 5.2.2)
>
> **W3 Limited experimental validation**
>
> We appreciate the reviewer raising this point. We have expanded our experiments in the multitask applications and transfer learning, to investigate how our model scales with the training size, the number of tasks, and the sequence length . In particular:
>
> * Scaling with the training size and compare invariant versus non-invariant baseline (Fig 9)
>   * We have increased the training dataset size and compared ASEN with its non-invariant baseline for the single-task learning setting. As shown in Figure 9, at larger scale ($r=3.0, 4.0$), ASEN (with the correct invariance) still outperforms its non-invariant counterpart across most tasks, and matches it for Detect Capital.
>
> * Scaling with the training size (Fig 6 - top) and the number of tasks (Fig 6 - bottom)
>   * We have added r=2.0, 3.0 training units to investigate scaling with the amount of training dataset. We find that with more data, eventually the single-task setting also achieves similar  performance as the multitask setting at convergence, albeit with slower convergence speed and higher variance.
>   * We have added n=4, 5, 6 number of tasks (per subplot) to show the effect of increasing task count. We find that while more tasks leads to better performance at low-data regime, such benefits diminish with increasing training data, suggesting a practical tradeoff between training size and task count.
>
> * Scaling with the sequence length
>   * In Sec 5.2.2’s transfer-learning application, we also vary the sequence length and find that the pretraining effect varies across sequence length and tasks, as reported in Figure 11. Increasing the sequence length leads to a larger symmetry group, but also introduces greater inter-group difference across tasks. Thus, we find that pretraining always helps, but the magnitude of its impact is not necessarily higher for longer sequences.
>
> In summary:
> * ASEN with multitask training provides advantages in a low-data regime, whereas ASEN with single-task training can be as competitive by allowing longer training time and larger dataset size.
> * That said, the synthetic sequence tasks we consider are all independent. If the tasks are related (for example, if an easier task could provide useful intermediate structure for solving a harder one), ASEN may offer advantages even at scale. Exploring this scenario is an interesting direction for future work.
>
> ---
>
> We thank the reviewer for helping improve the quality of our paper, and remain available to address any further questions or concerns.

---

### Official Review · Reviewer_sRQM · 2025-11-01

**Soundness:** 3
**Presentation:** 2
**Contribution:** 1
**Rating:** 2
**Confidence:** 3

**Summary:**

The paper presents a method to obtain foundational models equivariant to any chosen subgroup of a larger symmetry group while reusing a single backbone. It introduces a fixed, symmetry-breaking hypergraph whose automorphisms match the target subgroup, so the backbone effectively enforces just the chosen symmetries. The authors provide proofs of this property and show that their models inherit universality from a universal base model. They then present experiments evaluating the framework on human pose estimation, traffic flow prediction, and image classification benchmarks, along with synthetic tests that explore multitask and transfer learning.

**Strengths:**

- The authors tackle a relevant problem: building foundation models equivariant with respect to adaptive symmetries.
- The proposed solution, using symmetry breaking to realize adaptable group equivariance, is a brilliant idea.

**Weaknesses:**

1. Although the proposed solution is interesting, the analysis is broad but shallow. The paper aims to span both theory and practice, yet it defers the hardest, and most relevant, questions to future work, even though they are essential to validate the framework as a practical path to adaptive foundation models. In particular:
	- The limited analysis of scalability and computational cost is a critical omission, and should be of paramount importance in the design of foundation models.
	- A thorough analysis of how the $k$-closure approximation affects expressivity, equivariance, and performance is fundamental to assessing the framework’s practical relevance and implementability.
2. Readability is limited: the introduction is technical from the introduction, the exposition is overly compact, and there are too few worked examples to illustrate the core mechanisms.

**Questions:**

1. How can the presented baselines be scaled to test the scalability potential of the proposed model?
2. In this direction, would it be possible to report scaling laws for this models: how do preprocessing and training costs grow with input size, the number of hypergraph automorphisms, and with $k$?
3. At scale, when does this approach help with respect to a non-equivariant baseline?
4. Which are the groups for which do you envision a particular difference in employing its $2$-closure?

---

> ### Author Response · Authors · 2025-11-24
>
> We thank the reviewer for their thoughtful assessment of our work, and their appreciation of our “brilliant idea”. However, *we respectfully disagree that our analysis is too shallow to validate our framework as a viable method for equivariant foundation models in the future*. Our key contribution is to establish the **conceptual soundness of ASEN for building flexible equivariant models**, from both theoretical and practical perspectives across diverse learning settings (including graphs, images, and sequences). We also provided empirical results on the scalability of our framework when scaling training dataset size: see Table 6 for comparing ASEN with non-equivariant counterparts on the single-task setting, and Figure 6 for the multitask application.
>
> To address the reviewer’s concerns, we have provided **an additional set of scalability experiments**, investigating our model’s scaling behavior with respect to training dataset size (revised Table 6 - top, Figure 9), the number of tasks (revised Table 6 - bottom), and the input sequence length (Figure 11). We have also reported the training compute costs (Figure 10). Please see our responses to concerns raised in the Weakness and Questions section below, and the revised paper for more details.
>
> ---
>
> **W1: Limited analysis of scalability**
>
> We thank the reviewers for raising this point. We agree that scalability and computational considerations are important for *future foundation-model–scale architectures*. However, we would like to clarify that designing a foundation model is not the goal of this paper. Our contribution is instead:
> * A general framework (ASEN) for building *flexible equivariant models via symmetry breaking*, supported by theoretical guarantees (e.g., achieve the correct subgroup symmetries and universality properties)
> * *Proof-of-concept empirical studies* demonstrating the flexibility and effectiveness of our framework (e.g., implement diverse symmetries and allow shared symmetry knowledge transfer)
>
> Our paper proposes ASEN as a conceptual and principled step towards equivariant foundation models — not a full-scale foundation model. That said, we agree that questions of scalability and computational efficiency become essential at large model scales, and we have expanded our empirical studies to address these. See responses to Q1-Q3 below.
>
> **Q1:  Scalability potential of our method**
>
> We have expanded our experiments in the multitask applications and transfer learning, to investigate how our model scales with the training size, the number of tasks, and the sequence length. In particular:
>
> * Scaling with the training size (Fig 6 - top) and the number of tasks (Fig 6 - bottom)
>   * We have added r=2.0, 3.0 training units to investigate scaling with the amount of training dataset. We find that with more data, eventually the single-task setting also achieves similar performance as the multitask setting at convergence, albeit with slower convergence speed and higher variance.
>   * We have added n=4, 5, 6 number of tasks (per subplot) to show the effect of increasing task count. We find that while more tasks leads to better performance at low-data regime, such benefits diminish with increasing training data, suggesting a practical tradeoff between training size and task count.
>
> * Scaling with the sequence length
>   * In Sec 5.2.2 transfer-learning application, we also vary the sequence length and find that the pretraining effect varies across sequence length and tasks, as reported in Figure 11. Increasing the sequence length leads to a larger symmetry group, but also introduces greater inter-group difference, resulting in a non-monotonic pretraining effect.
>
> In summary:
> * ASEN with multitask training provides advantages in a low-data regime, whereas ASEN with single-task training can be as competitive by allowing longer training time and larger dataset size.
> * That said, the synthetic sequence tasks we consider are all independent. If the tasks are related (for example, if an easier task could provide useful intermediate structure for solving a harder one), ASEN may offer advantages even at scale. Exploring this scenario is an interesting direction for future work.

---

> > ### Author Response · Authors · 2025-11-24
> >
> > **Q2: Computation costs**
> >
> > We thank the reviewer for raising this important question. For our implementation with $K=2$: the preprocessing cost (Algorithm 1) scales as $O(r n^2)$ where $n$ is the number of nodes and $r$ is the number of the generators of the group $G$. The training cost with a message-passing neural network backbone scales as $O(m n^2)$ on a fully-connected dense graph where $m$ is the number of training data points, and $O(m |E|)$ on a sparse graph where $|E|$ is the number of edges. For general $K \geq 2$, the preprocessing cost scales as $O(r n^K)$ and the training cost scales as $O(m n^K)$ for the dense graph.
> > In practice, the preprocessing step is performed only once before training, so its cost is negligible compared to the training cost. We have also reported the training cost as a function of the training data size and the number of tasks in Figure 10.
> >
> > **Q3: Comparison with non-equivariant baseline at scale**
> >
> > - We have further increased the training dataset size and compared ASEN with its non-invariant baseline for the single-task learning setting. As shown in Figure 9, at larger scale ($r=3.0, 4.0$), ASEN (with the correct invariance) still outperforms its non-invariant counterpart across most tasks, and matches it for Detect Capital.
> > - There is a large body of theoretical works studying the generalization benefits of equivariant models, see for example [3,4]. There are also empirical studies on equivariance at scale, such as [5]. We agree with the reviewer that investigating the role of equivariance at scale is an important direction. However, this topic requires a dedicated research effort of its own, and pursuing it is beyond the scope of the present work.
> >
> > **W1: Limited analysis of 2-closure approximation**
> >
> > We have added more discussions on related works characterizing the permutation subgroups being equal to their 2-closure group, and the approximation-generalization tradeoff when choosing the incorrect symmetry groups. See line 177-182 in the revised paper for full details. In particular:
> > * Many permutation subgroups are equal to their 2-closure groups independent of the permutation representations (known as totally 2-closed). These include ﬁnite nilpotent groups that are either cyclic or a direct product of a generalized quaternion group with a cyclic group of odd order [1]. There are extensive studies and active research in group theory to characterize groups that are totally 2-closed (see references in [1]).
> > * When the group is different from its 2-closure group, this can be viewed as choosing an incorrect symmetry group larger than the ground truth. We can use results from [2] to obtain the approximation-generalization tradeoff, where the larger incorrect symmetry group introduces bias while reducing variance.
> >
> > **W2: Limited readability**
> >
> > We added more discussions on the 2-closure approximation (line 177-182), the practical relevance of Theorem 2 (line 277-282), the computational costs of our framework (line 299-302), and more scaling experiments to illustrate our framework (e.g., updated Fig.6, Sec 5.2.2).
> >
> > **Q4: Groups that differ from their $k$-closure groups**
> >
> > We have already included a negative example where the 2-closure of $G$ is not equal to $G$, for $G=A_n$ the alternating group; see Appendix C.5. Other examples where $G^{(2)} > G$ include a direct product of a generalized quaternion group with a cyclic group of even order [1].
> >
> > ---
> >
> > We thank the reviewer for helping improve the quality of our paper, and remain available to address any further questions or concerns.
> >
> > References
> >
> > 1. Abdollahi and Arezoomand, Finite nilpotent groups that coincide with their 2-closures in all of their faithful permutation representations, J. Algebra Appl. 17(4)(2018) 1850065.
> > 2. Huang, Levie, & Villar, Approximately equivariant graph networks. NeurIPS 2023.
> > 3. Elesedy & Zaidi, Provably strict generalisation benefit for equivariant models. ICML 2021.
> > 4. Bietti et al., On the sample complexity of learning under geometric stability. NeurIPS 2021.
> > 5. Brehmer et al., Does equivariance matter at scale?. TMLR 2025.

---

### Official Review · Reviewer_8QJx · 2025-11-01

**Soundness:** 3
**Presentation:** 4
**Contribution:** 3
**Rating:** 8
**Confidence:** 4

**Summary:**

This paper presents a new method for constructing equivariant neural networks by first having a base network equivariant to a larger base group and then fixing a symmetry-breaking input feature which is precisely invariant with respect to the target (sub)group. The authors focus on the specific case of base group being Sn, for which the authors draw upon a classical result that one can always construct a hypergraph of which automorphism group is equal to the target group and propose an approximate scheme that restricts the search space to graphs. The authors prove theoretical sanity-checks on equivariance and expressive power. Through experiments, the authors show that the proposed method can use different symmetries under one architecture and provide a diagnostic insight into the choice of the target group structure, and also show evidence that the proposed method exhibits knowledge sharing and transfer across tasks with different symmetries.

**Strengths:**

S1. While there has been prior work on architecture agnostic invariance/equivariance and knowledge sharing/transfer across symmetries [1-3], they consider taking an unconstrained base model and adding symmetry constraints through (randomized) symmetry breaking, while this work considers an opposite and original direction of starting at an overly constrained base model and reducing symmetry constraints using a fixed symmetry-breaking input. This is an interesting direction and could facilitate future work.

S2. The paper is very well written and easy to follow.

S3. The proposed method is simple and technically sound as far as I can confirm.

S4. The experimental results support the claims made in the main text.

S5. The shown connections to hypergraphs and higher-order GNNs are interesting and might lead to renewed interests of related literature.

[1] Kim et al., Learning Probabilistic Symmetrization for Architecture Agnostic Equivariance, NeurIPS 2023.

[2] Mondal et al., Equivariant Adaptation of Large Pretrained Models, NeurIPS 2023.

[3] Kim et al., Revisiting Random Walks for Learning on Graphs, ICLR 2025.

**Weaknesses:**

W1. There are several potential weaknesses stemming from the fact that the work takes an over-constrained base model and relaxes its constraints in downstream tasks. One weaknesses is related to practical implication of Theorem 2. For large groups G such as Sn as considered in this work, constructing universal and equivariant networks is in general quite hard and requires combinatorially large feature spaces [4] or some kind of randomized symmetry breaking themselves [5], making the setup of Theorem 2 unlikely in practice. Another potential weaknesses is that, with the base model overly constrained, the scope of knowledge it can accumulate from pretraining without knowing what downstream tasks are can be substantially limited. As far as I understand, pretraining in Section 5.2.2 is aware of downstream tasks and uses task-specific embedding modules, so it bypasses this problem; but in general, one may not know the details of downstream tasks during pretraining.

[4] Maron et al., On the Universality of Invariant Networks, ICML 2019.

[5] Abboud et al., The Surprising Power of Graph Neural Networks with Random Node Initialization, IJCAI 2021.

**Questions:**

I don't have particular questions but would like to hear the authors' opinions on W1.

---

> ### Author Response · Authors · 2025-11-24
>
> We thank the reviewer for their encouraging feedback, and their appreciation of the *novelty, impactfulness, soundness, and clarity* of our paper. We provide detailed responses to concerns raised in the Weakness section.
>
> **W1. Practical relevance of Theorem 2 (Universality of ASEN follows from Universality of its base model)**
>
> We agree with the reviewer that for $S_n$ on graphs (or symmetric matrices) by the conjugation action, the universality result typically requires combinatorically large feature space or can only be guaranteed in a probabilistic sense. However, for graphs with distinct node features, message-passing graph neural networks (used as our backbone model) are universal [1], and thus Theorem 2 can be applied.
> Moreover, for $S_n$ acting on sets or points, there exist universal architectures with polynomial complexity, see for example [2,3].
> We have added a remark after Theorem 2 to discuss its practical relevance (line 276-282).
>
> **W2. Limited pretraining effect without knowing downstream task details**
>
> * We appreciate the reviewer raising this point, and would like to clarify our set-up: In Section 5.2.2, we only assume knowledge of the downstream task *domain* (i.e., on a sequence), without the concrete task details. In theory, one only needs to know the downstream task domains, and a sufficiently large group which contains the downstream tasks’ symmetry groups as subgroups. In the experiment, we pretrain the model on task 1, task 2 (unaware of task 3 details), and then finetune the pretrained model in task 3.
> * We do acknowledge that our method requires the knowledge of the task domain (and thereby exploiting the domain-related symmetries such as graph automorphisms or sequence reversals). Extending our framework to multi-modal settings (without knowing the downstream task domain) is an interesting future direction. We have added these discussions in Sec.6 (line 517-520).
>
> We thank the reviewer for helping improve the quality of our paper, and remain available to address any further questions or concerns.
>
> References:
>
> 1. Louka, What graph neural networks cannot learn: depth vs width. ICLR 2020
> 2. Hordan et al., Complete neural networks for complete euclidean graphs. AAAI 2024
> 3. Blum-Smith et al., A Galois theorem for machine learning: Functions on symmetric matrices and point clouds via lightweight invariant features. ArXiv 2405.08097.

---

> > ### Comment · Reviewer_8QJx · 2025-11-24
> >
> > Thank you for the thoughtful response. On W1, assuming unique node features, or only requiring set-universal functions, does not completely remedy this problem (if I understand correctly), as the former does not hold for e.g. the considered tasks in Table 3, and the authors are taking $V$ as a space of (hyper)graphs so Theorem 2 essentially needs universality of (hyper)graph functions. So I would remain that Theorem 2 has limited practical relevance for the algorithm presented in the paper, although I can see its usefulness in generally understanding equivariant functions under the proposed symmetry breakings. I would advise the authors to make this point clear (if my argument is correct).
> >
> > On W2, to be clear, am I correct in understanding that: for transfer learning, one has to (1) know in advance the symmetries of all downstream tasks (or a group having all of them as subgroups) that's going to be tested (while not the tasks themselves), and (2) have a pretraining dataset that is a mixture of some of the downstream tasks with known symmetries? I find this setup interesting, but for general audience, would it be possible for the authors to provide practical applications or problems where this pretraining-transfer could be useful and impactful, compared to current methods used in these applications?

---

> > > ### Author Response · Authors · 2025-11-25
> > >
> > > We thank the reviewer for engaging with our rebuttal and the follow-up questions.
> > >
> > > **On W1**
> > >
> > > We thank the reviewer for recognizing the general usefulness of Theorem 2. The reviewer is correct to say the universality condition doesn’t apply to the sequence tasks where node features take values from a fixed vocabulary (Table 3.). However, Theorem 2 *does* apply to the graph tasks (Table 1, 2) where the node features are real-valued signals representing joint coordinates or traffic volume, and therefore are distinct with high probability. We will clarify this further in the paper revision.
> > >
> > > **On W2**
> > >
> > > * Your understanding is mostly correct, but we clarify two subtle points:
> > >   * In (1), knowing a group $G$ having all downstream task symmetries as subgroups is generally *weaker* than knowing the exact symmetries of all downstream tasks; our framework requires only this weaker condition.
> > >   * Moreover, we may relax the requirement from *all* downstream tasks to *most* downstream tasks, allowing some tasks having symmetry group $G’ \neq G$, so long as $G’$ is sufficiently “close” to $G$.
> > > * Great question! One impactful application is on **multiple-physics pretraining**: current methods [1,2] demonstrated the ability of a foundation model — pretrained on diverse PDEs across domains — can generalize or transfer to new PDEs. However, these methods use the same symmetry for all tasks (e.g., translation invariance for 2D regular grid [1], permutation invariance for arbitrary geometries [2]), **without accounting for task-specific symmetry breaking**, such as boundary conditions or material heterogeneity. Our framework ASEN can adapt to the specific symmetries of each PDE with a common backbone model, potentially yielding a more expressive and transferable multi-physics foundation model.
> > >
> > > References
> > >
> > > 1. McCabe et al., Multiple physics pretraining for spatiotemporal surrogate models, NeurIPS 2024
> > > 2. Rahman et al., Pretraining codomain attention neural operators for solving multiphysics PDEs, NeurIPS 2024

---

> > > > ### Comment · Reviewer_8QJx · 2025-11-26
> > > >
> > > > Thank you for the follow-up. After reading the responses and other reviews, my assessment is that this is a technically solid work with good contributions and presentations, with initially raised limitations remaining to some degree (Theorem 2 requires assumptions sometimes not holding in practice, and transfer learning setup requires knowledges of downstream task domains) but with also clear directions for improvements as proposed in the responses. Therefore I would like to retain my supportive rating and recommend acceptance.

---

> > > > > ### Author Response · Authors · 2025-11-26
> > > > >
> > > > > Thank you again for reading our follow-up responses, and for recommending our paper for acceptance. We will incorporate the discussions on Theorem 2, transfer learning set-up, and future application directions into the revised version of the paper.

---

### Author Response · Authors · 2025-12-01

Dear AC and reviewers,

Our work introduces Any-Subgroup Equivariant Networks (ASEN), a general framework for building *flexible* equivariant models via symmetry breaking. Our framework is supported by theoretical guarantees (e.g., achieve the correct subgroup symmetries, connections to equivariant MLPs, and universality properties), as well as proof-of-concept empirical studies on its flexibility and effectiveness (e.g., implement diverse symmetries and allow shared symmetry knowledge transfer).

*Initial reviews* reflected **broad recognition of our contribution**: all reviewers except one recommended *accept, good paper*; we believe the lower score from Reviewer sRQM reflects a conceptual misunderstanding of the paper’s intended purpose and scope. The main concerns raised were:
1. Scalability and computational considerations (sRQM, gDLb)
2. 2-closure approximation (sRQM, VJcc)
3. Practical applicability of Theorem 2 and transfer learning set-up (8QJx)
4. Experimental and conceptual comparisons to equivariant MLPs (VJcc)

*Our rebuttal addressed these concerns as follows*:

1. Scalability considerations
* *Clarification of our goal and scope*: our paper proposes ASEN as a conceptual and principled step towards equivariant foundation models — not a full-scale foundation model, as Reviewer sRQM misinterpreted. We agree that scalability and computational efficiency become essential at large model scales, and thus have provided more experiments to address these considerations (see next point).

* *Additional scalability experiments and compute information*: we investigated our framework’s scaling behavior with respect to training dataset size (revised Table 6 - top, Figure 9), the number of tasks (revised Table 6 - bottom), and the input sequence length (Figure 11); we have also discussed the computational complexity (line 299-303) and reported the training compute costs (Figure 10).

2. *Additional Discussion on the 2-closure approximation*: we discussed related works in characterizing classes of permutation subgroups being *equal to* their 2-closure group (i.e., no approximation), and how to analyze the approximation-generalization tradeoff when choosing the incorrect symmetry groups (177-182).

3. Theorem 2; transfer-learning

* *Clarifications of Theorem 2 on its general usefulness and applications*: we clarified that Theorem 2 states that whatever variety of universality that the base model enjoys is inherited by ASEN; this applies to existing architectures for point clouds and message-passing architectures for graphs assuming unique node features (e.g., in our graph task experiments Sec 5.1); see line 277-282.

* *Discussions on the transfer-learning set-up and its applications*: we explained that our set-up only requires knowing a group $G$ having most downstream task symmetries as subgroups, and a pretraining dataset as a mixture of some downstream tasks with known symmetries; we discussed extending this set-up to multiphysics pretraining is a promising future direction.

4. *Connections to equivariant MLPs*: we clarified that we have already provided experimental comparison with equivariant MLPs (i.e., the baseline from Huang et al.) in Sec 5.1; we emphasized that ASEN can benefit from cross-task symmetry transfer, whereas equivariant MLPs cannot (with empirical evidence in Sec 5.2.)

*Positive rebuttal outcome*: Both reviewers 8QJx and VJcc actively engaged in the rebuttal and maintained their initial supportive ratings, recommending the acceptance of our paper. We believe our work offers a valuable framework for building flexible equivariant models and paving the way towards large-scale equivariant foundation models.

---

### Meta-Review · Area_Chair_Xwbe · 2026-01-10

**Summary:**

Three reviewers recommended acceptance, appreciating the novel idea of symmetry breaking and the solid mathematical proofs, and one reviewer (sRQM) gave rejection, raising the issues of shallow analyses and unclear writing, in particular, that of scalability. The authors’ rebuttal addresses most of the reviewers’ questions and concerns, providing detailed discussions and additional experiments. AC finds that the authors’ responses are convincing and this paper is worth being shared in the community, recommending acceptance.

**Reviewer Concerns:**

The rebuttal clarified missing details and also provided additional analyses and strong results.
The issue of demonstrating the performance of multi-task learning on real datasets (reviewr VJcc ) remains, but AC finds this is not critical.

**Reviewer Scores:**

Reviewer 8QJx retained the original rating of 8 after discussion.
Reviewer sRQM would have changed the score from 2 to 6.
Reviewer gDLb would have retained the original score 8.
Reviewer VJcc retained the original rating of 8 after discussion.

---

### Decision · Program_Chairs · 2026-01-26

Accept (Poster)